# Global Rewards in Restless Multi-Armed Bandits

**Naveen Raman**
Carnegie Mellon University
naveenr@cmu.edu

**Zheyuan Ryan Shi**
University of Pittsburgh
ryanshi@pitt.edu

**Fei Fang**
Carnegie Mellon University
feifang@cmu.edu

## Abstract

Restless multi-armed bandits (RMAB) extend multi-armed bandits so pulling an arm impacts future states. Despite the success of RMABs, a key limiting assumption is the separability of rewards into a sum across arms. We address this deficiency by proposing restless-multi-armed bandit with global rewards (RMAB-G), a generalization of RMABs to global non-separable rewards. To solve RMAB-G, we develop the Linear- and Shapley-Whittle indices, which extend Whittle indices from RMABs to RMAB-Gs. We prove approximation bounds but also point out how these indices could fail when reward functions are highly non-linear. To overcome this, we propose two sets of adaptive policies: the first computes indices iteratively, and the second combines indices with Monte-Carlo Tree Search (MCTS). Empirically, we demonstrate that our proposed policies outperform baselines and index-based policies with synthetic data and real-world data from food rescue.

## 1 Introduction

Restless multi-armed bandits (RMAB) are models of resource allocation that combine multi-armed bandits with states for each bandit's arm. Such a model is "restless" because arms can change state even when not pulled. RMABs are an enticing framework because optimal decisions can be efficiently made using pre-computed Whittle indices [1, 2]. As a result, RMAB have been used in scheduling [3], autonomous driving [4], multichannel access [5], and maternal health [6].

The existing RMAB model assumes that rewards are separable into a sum of per-arm rewards. However, many real-world objectives are non-separable functions of the arms pulled. We find this situation in food rescue platforms. These platforms notify subsets of volunteers about upcoming food rescue trips and aim to maximize the trip completion rate [7]. When modeling this with RMABs, arms correspond to volunteers, arm pulls correspond to notifications, and the reward corresponds to the trip completion rate. The trip completion rate is non-separable as we only need one volunteer to complete each trip (more details in Section 3). Beyond food rescue, RMABs can potentially model problems in peer review [8], blood donation [9], and emergency dispatch [10], but the rewards are again non-separable.

Motivated by these situations, we propose the restless multi-armed bandit with global rewards (RMAB-G), a generalization of RMABs to non-separable rewards. Whittle indices from RMABs [1] fail for RMAB-Gs due to the non-separability of the reward. Because of this, we propose a generalization of Whittle indices to global rewards called the Linear-Whittle and Shapley-Whittle indices. Our approximation bounds demonstrate that these policies perform well for near-linear rewards but struggle for highly non-linear rewards. We address this by developing adaptive policies that combine

38th Conference on Neural Information Processing Systems (NeurIPS 2024).

Linear- and Shapley-Whittle indices with search techniques. We empirically verify that adaptive policies outperform index-based and baseline approaches on synthetic and food rescue datasets. [1]

Our contributions are: (1) We propose the RMAB-G problem, which extends RMABs to situations with global rewards. We additionally characterize the difficulty of solving and approximating RMAB-Gs; (2) We develop the Linear-Whittle and Shapley-Whittle indices, which extend Whittle indices for global rewards, and detail approximation bounds; (3) We design a set of adaptive policies which combine Linear- and Shapley-Whittle indices with greedy and Monte Carlo Tree Search (MCTS) algorithms; and (4) We empirically demonstrate that adaptive policies improve upon baselines and pre-computed index-based policies for the RMAB-G problem with synthetic and food rescue datasets.

## 2 Background and Notation

An instance of restless multi-armed bandits (RMABs) consists of $N$ arms, each of which is defined through the following Markov Decision Process (MDP): $(\mathcal{S}, \mathcal{A}, R_i, P_i, \gamma)$. Here, $\mathcal{S}$ is the state space, $\mathcal{A}$ is the action space, $R_i : \mathcal{S} \times \mathcal{A} \to \mathbb{R}$ is a reward function, $P_i : \mathcal{S} \times \mathcal{A} \times \mathcal{S} \to [0, 1]$ is a transition function detailing the probability of an arm transitioning into a new state, and $\gamma \in [0, 1)$ is a discount factor. For a time period $t$, we define the state of all arm as $\mathbf{s}^{(t)} = \{s_1^{(t)}, \cdots, s_N^{(t)}\}, s_i^{(t)} \in \mathcal{S}$, and for each round, a planner takes action $\mathbf{a}^{(t)} = \{a_1^{(t)}, \cdots, a_N^{(t)}\}, a_i^{(t)} \in \mathcal{A}$; we drop the superscript $t$ when clear from context. Following prior work [11, 12], we assume $\mathcal{A} = \{0, 1\}$ throughout this paper to represent not pulling and pulling an arm respectively. A planner chooses $\mathbf{a}^{(t)}$ subject to a budget $K$, so that at most $K$ arms can be pulled in each time step: $\sum_{i=1}^{N} a_i^{(t)} \leq K, \forall t$. The objective is to find a policy, $\pi : \mathcal{S}^N \to \mathcal{A}^N$ that maximizes the discounted total reward, summed over all arms:

$$\max_{\pi} \mathbb{E}_{(\mathbf{s},\mathbf{a}) \sim (P,\pi)} \left[ \sum_{t=0}^{\infty} \sum_{i=1}^{N} \gamma^t R_i(s_i^{(t)}, a_i^{(t)}) \right] \tag{1}$$

To solve an RMAB, one can pre-compute the "Whittle index" for each arm $i$ and possible state $s_i$: $w_i(s_i) = \min_{w} \{w | Q_{i,w}(s_i, 0) = Q_{i,w}(s_i, 1)\}$, where $Q_{i,w}(s_i, a_i) = -wa_i + R_i(s_i, a_i) + \gamma \sum_{s'} P_i(s_i, a_i, s') V_{i,w}(s')$, and $V_{i,w}(s') = \max_{a} Q_{i,w}(s', a)$. Here, $Q_{i,w}(s_i, a_i)$ represents the expected future reward for playing action $a_i$, given a penalty $w$ for pulling an arm. The Whittle index computes the minimum penalty needed to prevent arm pulls; larger penalties ($w_i(s_i)$) indicate more value for pulling an arm. In each round, the planner pulls the arms with the $K$ largest Whittle indices given the states of the arms. Such a Whittle index-based policy is asymptotically optimal [1, 2].

## 3 Defining Restless Multi-Armed Bandits with Global Rewards

Objectives in RMABs are written as the sum of per-arm rewards (Equation 1), while real-world objectives involve non-separable rewards. For example, food rescue organizations, which notify volunteers about food rescue trips, aim to maximize the trip completion rate, i.e., the fraction of completed trips. Suppose we model volunteers as arms and volunteer notifications as arm pulls. Then arm states correspond to volunteer engagement (e.g., active or inactive) and rewards are the probability that any notified volunteers complete the rescue. Here, we cannot split the reward into per-volunteer functions.

Inspired by situations such as food rescue, we propose the restless multi-armed bandit with global rewards (RMAB-G) problem, which generalizes RMABs to problems with a global reward function:

**Definition 1.** *We define an instance of* RMAB-G *through the following MDP for each arm:* $(\mathcal{S}, \mathcal{A}, R_i, R_{\mathrm{glob}}, P_i, \gamma)$. *The aim is to find a policy,* $\pi : \mathcal{S}^N \to \mathcal{A}^N$, *that maximizes:*

$$\max_{\pi} \mathbb{E}_{(\mathbf{s},\mathbf{a}) \sim (P,\pi)} \left[ \sum_{t=0}^{\infty} \gamma^t \left( R_{\mathrm{glob}}(\mathbf{s}^{(t)}, \mathbf{a}^{(t)}) + \sum_{i=1}^{N} R_i(s_i^{(t)}, a_i^{(t)}) \right) \right] \tag{2}$$

Throughout this paper, we focus on scenarios where $R_{\mathrm{glob}}(\mathbf{s}, \mathbf{a})$ is monotonic and submodular in $\mathbf{a}$; $R_{\mathrm{glob}}(\mathbf{s}, \mathbf{a}) = F_{\mathbf{s}}(\{i | a_i = 1\})$, where $F$ is submodular and monotonic. Such rewards have

---

diminishing returns as extra arms are pulled. We select this because a) monotonic and submodular functions can be efficiently optimized [13] and b) submodular functions are ubiquitous [14, 15, 8].

Our problem formulation can model various real-world scenarios including notification decisions in food rescue and reviewer assignments in peer review [8]. In food rescue, let $p_i(s_i)$ be the probability that a volunteer picks up a given trip, given their state, $s_i$. The global reward is the probability that any notified volunteer accepts a trip: $R_{\text{glob}}(\mathbf{s}, \mathbf{a}) = 1 - \prod_{i=1}^{N}(1 - a_i p_i(s_i))$. In peer review, journals select reviewers for submissions where selection impacts future reviewer availability [8]. The goal is to select a subset of available reviewers with comprehensive subject-matter expertise. If reviewer expertise is a set $Y_i$, then we maximize the overlap between the required expertise, $Z$, and the expertise across selected ($a_i = 1$) and available ($s_i = 1$) reviewers: $R_{\text{glob}}(\mathbf{s}, \mathbf{a}) = |\bigcup_{i, s_i=1, a_i=1} Y_i \cap Z|$. Our formulation inherits the per-arm reward $R_i$ from RMABs as there may be per-arm rewards. For example, food rescue platforms might care about the number of active volunteers, so $R_i(s_i, a_i) = s_i$.

Let $R(\mathbf{s}, \mathbf{a}) = R_{\text{glob}}(\mathbf{s}, \mathbf{a}) + \sum_{i=1}^{N} R_i(s_i, a_i)$. We characterize the difficulty of RMAB-G:

**Theorem 1.** *The restless multi-armed bandit with global rewards is PSPACE-Hard.*

**Theorem 2.** *No polynomial time algorithm can achieve better than a $1 - 1/e$ approximation to the restless multi-armed bandit with global rewards problem.*

**Theorem 3.** *Consider an RMAB-G instance $(\mathcal{S}, \mathcal{A}, R_i, R_{\text{glob}}, P_i, \gamma)$. Let $Q_\pi(\mathbf{s}, \mathbf{a}) = \mathbb{E}_{(\mathbf{s}', \mathbf{a}') \sim (\mathcal{P}, \pi)}[\sum_{t=0}^{\infty} \gamma^t R(\mathbf{s}', \mathbf{a}'))|\mathbf{s}^{(0)} = \mathbf{s}, \mathbf{a}^{(0)} = \mathbf{a}]$ and let $\pi^*$ be the optimal policy. Let $\mathbf{a}^* = \arg\max_{\mathbf{a}} Q_{\pi^*}(\mathbf{s}, \mathbf{a})$. Suppose $P_i(s, 1, 1) \geq P_i(s, 0, 1) \forall s, i$, $P_i(1, a, 1) \geq P_i(0, a, 1) \forall i, a$, $R(\mathbf{s}, \mathbf{a})$ monotonic and submodular in $\mathbf{s}$ and $\mathbf{a}$, and $g(\mathbf{s}) = \max_{\mathbf{a}} R(\mathbf{s}, \mathbf{a})$ submodular in $\mathbf{s}$. Then, with oracle access to $Q_{\pi^*}$ we can compute $\hat{\mathbf{a}}$ in $\mathcal{O}(N^2)$ time so $Q_{\pi^*}(\mathbf{s}, \hat{\mathbf{a}}) \geq (1 - \frac{1}{e})Q_{\pi^*}(\mathbf{s}, \mathbf{a}^*)$.*

Theorem 1 and 2 demonstrate the difficulty of finding solutions to RMAB-Gs while Theorem 3 describes a potential panacea via $Q_{\pi^*}$. Leveraging $Q_{\pi^*}$ is a polynomial-time algorithm, but the exponential state space makes it difficult to learn such a $Q$, motivating the need for better algorithms (we empirically demonstrate this in Section 6.1). In essence, Lemma 3 demonstrates that it is possible to achieve a $1 - \frac{1}{e}$ approximation for the Q-value; however, even computing such an approximation is difficult, and motivates the need for computationally feasible solutions. Our assumptions for Lemma 3 essentially capture the idea that rewards are submodular in both the state $\mathbf{s}$ and action $\mathbf{a}$; these assumptions are satisfied for all of our synthetic experiments in Section 6. We prove Theorem 1 through reduction from RMAB, Theorem 2 through reduction from submodular optimization, and Theorem 3 through induction on value iteration. Full proofs are in Appendix K.

## 4 Linear-Whittle and Shapley-Whittle Indices

### 4.1 Defining Linear-Whittle and Shapley-Whittle

Because the Whittle index policy is asymptotically optimal for RMABs [2], we investigate its extension to RMAB-Gs. We develop two extensions: the Linear-Whittle policy, which computes marginal rewards for each arm, and the Shapley-Whittle policy, which computes Shapley values for each arm. To define Linear-Whittle, we let the marginal reward be $p_i(s_i) = \max_{\mathbf{s}'|s_i'=s_i} R_{\text{glob}}(\mathbf{s}', \mathbf{e}_i)$, where $\mathbf{e}_i$ is a vector with 1 at index $i$ and 0 elsewhere. The Linear-Whittle policy linearizes the global reward by optimistically estimating $R_{\text{glob}}(\mathbf{s}, \mathbf{a}) \approx \sum_{i=1}^{N} p_i(s_i) a_i$ and computing the Whittle index from this:

**Definition 2.** *Linear-Whittle - Consider an instance of RMAB-G $(\mathcal{S}, \mathcal{A}, R_i, R_{\text{glob}}, P_i, \gamma)$. Define $Q_{i,w}^L(s_i, a_i, p_i) = -a_i w + R_i(s_i, a_i) + a_i p_i(s_i) + \gamma \sum_{s_i'} P_i(s_i, a_i, s_i') V_{i,w}(s_i')$, where $V_{i,w}(s_i') = \max_{a_i'} Q_{i,w}^L(s_i', a_i', p_i)$. Then the Linear-Whittle index is $w_i^L(s_i, p_i) = \min_{w}\{w|Q_{i,w}^L(s_i, 0, p_i) > Q_{i,w}^L(s_i, 1, p_i)\}$. The Linear-Whittle policy pulls arms with the $K$ highest Linear-Whittle indices.*

To improve on the Linear-Whittle policy, we develop the Shapley-Whittle policy, which uses Shapley values to linearize the global reward. The Linear-Whittle policy linearizes global rewards via $p_i(s_i)$, but this approximation could be loose as $\sum_{i=1}^{N} p_i(s_i) a_i \geq R_{\text{glob}}(\mathbf{s}, \mathbf{a})$. The Shapley value, $u_i(s_i)$, improves this by averaging marginal rewards across subsets of arms (see Shapley et al. [16]). Let $\vee$

be the element-wise maximum, so $\mathbf{a} \vee \mathbf{e}_i$ means arms in $\mathbf{a}$ and arm $i$ are pulled. Define:

$$u_i(s_i) = \min_{\mathbf{s}'|s_i'=s_i} \sum_{\mathbf{a}\in\{0,1\}^N, a_i=0, \|\mathbf{a}\|_1\leq K-1} \frac{\|\mathbf{a}\|_1!(N-\|\mathbf{a}\|_1-1)!}{\frac{N!}{(N-K)!}} (R_{\mathrm{glob}}(\mathbf{s}', \mathbf{a}\vee\mathbf{e}_i) - R_{\mathrm{glob}}(\mathbf{s}', \mathbf{a})) \tag{3}$$

Here, $R_{\mathrm{glob}}(\mathbf{s}', \mathbf{a}\vee\mathbf{e}_i) - R_{\mathrm{glob}}(\mathbf{s}', \mathbf{a})$ captures the added value of pulling arm $i$, when $\mathbf{a}$ are already pulled. While the Linear-Whittle policy estimates the global reward as $R_{\mathrm{glob}}(\mathbf{s}, \mathbf{a}) \approx \sum_{i=1}^N p_i(s_i)a_i$, the Shapley-Whittle policy estimates the global reward as $R_{\mathrm{glob}}(\mathbf{s}, \mathbf{a}) \approx \sum_{i=1}^N u_i(s_i)a_i$. The Linear-Whittle policy overestimates marginal contributions, as $\sum_{i=1}^N p_i(s_i)a_i \geq R_{\mathrm{glob}}(\mathbf{s}, \mathbf{a})$ (proof in Appendix L), so by using Shapley values, we take a pessimistic approach (by taking the minimum over all $\mathbf{s}'$). This approach could lead to more accurate estimates of $R_{\mathrm{glob}}(\mathbf{s}, \mathbf{a})$ because marginal contributions are averaged across many combinations of arms:

**Definition 3.** *Shapley-Whittle - Consider an instance of* RMAB-G $(\mathcal{S}, \mathcal{A}, R_i, R_{\mathrm{glob}}, P_i, \gamma)$. *Let* $Q_{i,w}^S(s_i, a_i, u_i) = -a_i w + R_i(s_i, a_i) + a_i u_i(s_i) + \gamma \sum_{s_i'} P_i(s_i, a_i, s_i') V_{i,w}(s_i')$, *where* $V_{i,w}(s_i') = \max_{a_i'} Q_{i,w}^S(s_i', a_i', u_i)$. *Then the Shapley-Whittle index is* $w_i^S(s_i, u_i) = \min_w \{w | Q_{i,w}^S(s_i, 0, u_i) > Q_{i,w}^S(s_i, 1, u_i)\}$. *The Shapley-Whittle policy pulls arms with the $K$ highest Shapley-Whittle indices.*

### 4.2 Approximation Bounds

To characterize the performance of our policies, we prove approximation bounds (proofs in Appendix L). We first define three properties that characterize situations when policies based on the Whittle index are asymptotically optimal; we use these characteristics to define when our Linear- and Shapley-Whittle policies perform well:

1. **Indexability** refers to a property of an RMAB where increasing the penalty for pulling arms, $w$, from 0 to $\infty$ monotonically changes the arms that are pulled from all the arms to none of the arms (see Akbarzadeh and Mahajan [17] for details)

2. **Irreducibility** is a property of a Markov Chain if every pair of states is visitable; that is, there exists no subset of sink states such that one cannot escape from these sink states

3. **Uniform Global Attractor** is a property of an RMAB, where the state distribution converges to the optimal state distribution (in terms of expected reward) from any initial state; for more discussion, see Gast et al. [18].

We define two Theorems which capture lower bounds on the performance of our policies:

**Theorem 4.** *For any fixed set of transitions, $\mathcal{P}$, let* $\mathrm{OPT} = \max_\pi \mathbb{E}_{(\mathbf{s},\mathbf{a})\sim(P,\pi)}[\frac{1}{T}\sum_{t=0}^{T-1} R(\mathbf{s}, \mathbf{a})]$ *for some $T$. For an* RMAB-G $(\mathcal{S}, \mathcal{A}, R_i, R_{\mathrm{glob}}, P_i, \gamma)$, *let* $R_i'(s_i, a_i) = R_i(s_i, a_i) + p_i(s_i)a_i$, *and let the induced linear* RMAB *be* $(\mathcal{S}, \mathcal{A}, R_i', P_i, \gamma)$. *Let $\pi_{\mathrm{linear}}$ be the Linear-Whittle policy, and let* $\mathrm{ALG} = \mathbb{E}_{(\mathbf{s},\mathbf{a})\sim(P,\pi_{\mathrm{linear}})}[\frac{1}{T}\sum_{t=0}^{T-1} R(\mathbf{s}, \mathbf{a})]$. *Define $\beta_{\mathrm{linear}}$ as*

$$\beta_{\mathrm{linear}} = \min_{\mathbf{s}\in\mathcal{S}^N, \mathbf{a}\in[0,1]^N, \|\mathbf{a}\|_1\leq K} \frac{R(\mathbf{s}, \mathbf{a})}{\sum_{i=1}^N (R_i(s_i, a_i) + p_i(s_i)a_i)} \tag{4}$$

*If the induced linear* RMAB *is irreducible and indexable with the uniform global attractor property, then $\mathrm{ALG} \geq \beta_{\mathrm{linear}}\mathrm{OPT}$ asymptotically in $N$ for any set of transitions, $\mathcal{P}$.*

**Theorem 5.** *For any fixed set of transitions, $\mathcal{P}$, let* $\mathrm{OPT} = \max_\pi \mathbb{E}_{(\mathbf{s},\mathbf{a})\sim(P,\pi)}[\frac{1}{T}\sum_{t=0}^{T-1} R(\mathbf{s}, \mathbf{a})]$ *for some $T$. For an* RMAB-G $(\mathcal{S}, \mathcal{A}, R_i, R_{\mathrm{glob}}, P_i, \gamma)$, *let* $R_i'(s_i, a_i) = R_i(s_i, a_i) + u_i(s_i)a_i$, *and let the induced Shapley* RMAB *be* $(\mathcal{S}, \mathcal{A}, R_i', P_i, \gamma)$. *Let $\pi_{\mathrm{shapley}}$ be the Shapley-Whittle policy, and let* $\mathrm{ALG} = \mathbb{E}_{(\mathbf{s},\mathbf{a})\sim(P,\pi_{\mathrm{shapley}})}[\frac{1}{T}\sum_{t=0}^{T-1} R(\mathbf{s}, \mathbf{a})]$. *Define $\beta_{\mathrm{shapley}}$ as*

$$\beta_{\mathrm{shapley}} = \frac{\min\limits_{\mathbf{s}\in\mathcal{S}^N, \mathbf{a}\in[0,1]^N, \|\mathbf{a}\|_1\leq K} \frac{R(\mathbf{s}, \mathbf{a})}{\sum_{i=1}^N (R_i(s_i, a_i) + u_i(s_i)a_i)}}{\max\limits_{\mathbf{s}\in\mathcal{S}^N, \mathbf{a}\in[0,1]^N, \|\mathbf{a}\|_1\leq K} \frac{R(\mathbf{s}, \mathbf{a})}{\sum_{i=1}^N (R_i(s_i, a_i) + u_i(s_i)a_i)}} \tag{5}$$

*If the induced Shapley* RMAB *is irreducible and indexable with the uniform global attractor property, then $\mathrm{ALG} \geq \beta_{\mathrm{shapley}}\mathrm{OPT}$ asymptotically in $N$ for any choice of transitions, $\mathcal{P}$.*

**Corollary 5.1.** *Let $\pi_{\text{linear}}$ be the Linear-Whittle policy,* $\text{ALG} = \mathbb{E}_{(\mathbf{s},\mathbf{a}) \sim (P, \pi_{\text{linear}})}[\frac{1}{T} \sum_{t=0}^{T-1} R(\mathbf{s}, \mathbf{a})]$ *and* $\text{OPT} = \max_{\pi} \mathbb{E}_{(\mathbf{s},\mathbf{a}) \sim (P, \pi)}[\frac{1}{T} \sum_{t=0}^{T-1} R(\mathbf{s}, \mathbf{a})]$ *for some $T$. Then* $\text{ALG} \geq \frac{\text{OPT}}{K}$ *for any $\mathcal{P}$.*

$\beta$ lower bounds policy performance; small $\beta$ does not guarantee poor performance but large $\beta$ guarantees good performance. For example, $\beta_{\text{linear}}$ and $\beta_{\text{shapley}}$ are near 1 for near-linear global rewards, so Linear- and Shapley-Whittle policies perform well in those situations. We prove all three Theorems by bounding the gap between the linear (or Shapley) approximation and $R(\mathbf{s}, \mathbf{a})$ (proofs in Appendix L). Our key insight is that we can bound the performance of Linear- and Shapely-Whittle policies using similarities between the global rewards and the rewards of an induced RMAB. We note that the assumptions made by all Theorems are the same assumptions needed for the optimality of Whittle indices [2]. We present an example of these lower bounds:

**Example 4.1.** *Consider an $N = 4, K = 2$ scenario for RMAB-G with $\mathcal{S} = \{0, 1\}$. Let $R_{\text{glob}}(\mathbf{s}, \mathbf{a}) = | \bigcup_{i | a_i = 1, s_i = 1} Y_i |$ and $R_i(s_i, a_i) = 0$. Here, $Y_i$ are sets corresponding to each arm, with $Y_1 = Y_2 = \{1, 2, 3\}$, $Y_3 = \{1, 2\}$ and $Y_4 = \{3, 4\}$. Let $s_i = 1$ for all $i$, and consider the Linear-Whittle policy. Next, note that $p_i(s_i) = R(s_i \mathbf{e}_i, \mathbf{e}_i)$, so $p_1(1) = p_2(1) = 3, p_3(1) = p_4(1) = 2$.*

*Now let $\mathbf{a} = [1, 1, 0, 0]$ and let $\mathbf{s} = [1, 1, 1, 1]$. Here, $R(\mathbf{s}, \mathbf{a}) = |Y_1 \cup Y_2| = 3$, while $\sum_{i=1}^{N} p_i(s_i) a_i = 6$. We note that we select such an $\mathbf{a}$ and $\mathbf{s}$ because this minimizes the lower bound from $\beta_{\text{linear}}$; other selections will not lead to the correct lower bound due to the minimum in the equation for $\beta_{\text{linear}}$. Therefore, $\beta_{\text{linear}} = \frac{3}{6} = \frac{1}{2}$, so the Linear-Whittle policy is a $\frac{1}{2}$ approximation to this problem.*

We describe upper bounds and apply these bounds to Example 4.1.

**Theorem 6.** *Let $\hat{\mathbf{a}}(\mathbf{s}) = \arg\max_{\mathbf{a} \in [0,1]^N, \|\mathbf{a}\|_1 \leq K} \sum_{i=1}^{N} (R_i(s_i, a_i) + p_i(s_i) a_i)$. For a set of transitions $\mathcal{P} = \{P_1, P_2, \cdots, P_N\}$, let $\text{OPT} = \max_{\pi} \mathbb{E}_{(\mathbf{s},\mathbf{a}) \sim (P, \pi)}[\sum_{t=0}^{\infty} \gamma^t R(\mathbf{s}, \mathbf{a}) | \mathbf{s}^{(0)}]$ and $\text{ALG} = \mathbb{E}_{(\mathbf{s},\mathbf{a}) \sim (P, \pi_{\text{linear}})}[\sum_{t=0}^{\infty} \gamma^t R(\mathbf{s}, \mathbf{a}) | \mathbf{s}^{(0)}]$. Define $\theta_{\text{linear}}$ as follows:*

$$\theta_{\text{linear}} = \min_{\mathbf{s} \in \mathcal{S}^N} \frac{R(\mathbf{s}, \hat{\mathbf{a}}(\mathbf{s}))}{\max_{\mathbf{a} \in [0,1]^N, \|\mathbf{a}\|_1 \leq K} R(\mathbf{s}, \mathbf{a})} \tag{6}$$

*Then there exists some transitions, $\mathcal{P}$, and initial state $\mathbf{s}^{(0)}$, so that $\text{ALG} \leq \theta_{\text{linear}} \text{OPT}$*

**Theorem 7.** *Let $\hat{\mathbf{a}}(\mathbf{s}) = \arg\max_{\mathbf{a} \in [0,1]^N, \|\mathbf{a}\|_1 \leq K} \sum_{i=1}^{N} (R_i(s_i, a_i) + u_i(s_i) a_i)$. For a set of transitions $\mathcal{P} = \{P_1, P_2, \cdots, P_N\}$ and initial state $\mathbf{s}^{(0)}$, let $\text{OPT} = \max_{\pi} \mathbb{E}_{(\mathbf{s},\mathbf{a}) \sim (P, \pi)}[\sum_{t=0}^{\infty} \gamma^t R(\mathbf{s}, \mathbf{a}) | \mathbf{s}^{(0)}]$ and $\text{ALG} = \mathbb{E}_{(\mathbf{s},\mathbf{a}) \sim (P, \pi_{\text{shapley}})}[\sum_{t=0}^{\infty} \gamma^t R(\mathbf{s}, \mathbf{a}) | \mathbf{s}^{(0)}]$. Let $\theta_{\text{shapley}}$ be:*

$$\theta_{\text{shapley}} = \min_{\mathbf{s}} \frac{R(\mathbf{s}, \hat{\mathbf{a}}(\mathbf{s}))}{\max_{\mathbf{a} \in [0,1]^N, \|\mathbf{a}\|_1 \leq K} R(\mathbf{s}, \mathbf{a})} \tag{7}$$

*Then there exists some transitions, $\mathcal{P}$ and initial state $\mathbf{s}^{(0)}$ so that $\text{ALG} \leq \theta_{\text{shapley}} \text{OPT}$*

**Example 4.2.** *Recall the situation from Example 4.1. Let $s_i = 1$ for all $i$, and consider the Linear-Whittle policy. Construct construct $P_i$, so that $P_i(s, a, 1) = 1 \forall i, s, a$ or in other words, $s_i^{(t)} = 1 \forall i, t$. The Linear-Whittle policy will pull arms 1 and 2, as $p_1(1) + p_2(1) \geq p_i(1) + p_j(1) \forall i, j$. This results in an action $\mathbf{a} = [1, 1, 0, 0]$, so that $R(\mathbf{s}, \mathbf{a}) = 3$, while the optimal action is $\mathbf{a}^* = [0, 0, 1, 1]$ with $R(\mathbf{s}, \mathbf{a}^*) = 4$. Computing $\theta_{\text{linear}}$ matches this upper bound of $\frac{3}{4}$ in this scenario.*

Our upper bounds demonstrate that, for any reward function, there exists a set of transitions where our policies perform a $\theta$-fraction worse than optimal. $\theta_{\text{linear}}$ is small when the optimal action diverges from the action taken by the Linear-Whittle policy (analogously for Shapley-Whittle). Such situations occur because policies fail to account for inter-arm interactions (e.g. Example 4.2). We present lower bounds using the average reward because Whittle optimality holds in that situation. Otherwise, we follow prior work, and focus on discounted reward [11] (for more discussion see Ghosh et al. [19])

# 5 Adaptive Approaches for RMAB-G

## 5.1 Limitations of Pre-Computed Index-Based Policies

Despite the approximation bounds given in Appendix L, Linear- and Shapley-Whittle policies can exhibit poor performance when upper bounds $\theta_{\text{linear}}$ and $\theta_{\text{shapley}}$ are small. We detail such a situation:

**Lemma 8.** *(Informal) Let $\pi \in \{\pi_{\text{linear}}, \pi_{\text{shapley}}\}$ be either the Linear or Shapley-Whittle policies. There exist problem settings where $\pi$ will achieve an approximation ratio no better than $\frac{1-\gamma}{1-\gamma^K}$.*

In general, pre-computed index-based policies perform poorly because of their inability to consider inter-arm interactions (proof in Appendix L). This approximation ratio matches lower bounds, $\lim_{\gamma \to 1} \frac{1-\gamma}{1-\gamma^K} = \frac{1}{K} = \beta_{\text{linear}} = \beta_{\text{shapley}}$, which motivates new policies which select arms adaptively.

## 5.2 Iterative Whittle Indices

Lemma 8 motivates the need for policies that select arms adaptively and go beyond pre-computed indices. Based on this, we propose two new policies: Iterative Linear-Whittle and Iterative Shapley-Whittle. Both policies compute current time step rewards, $r_i$, based on the arms pulled, $X$.

**Definition 4.** *Iterative Linear-Whittle - Consider an instance of* RMAB-G $(\mathcal{S}, \mathcal{A}, R_i, R_{\text{glob}}, P_i, \gamma)$. *Consider a state $\mathbf{s}$, and construct a new state, $\bar{\mathbf{s}}$, with $P_{\bar{s}_i} = P_i$. Let $Q_{i,w}^{IL}(\bar{s}_i, a_i, p_i, r_i^L, X) = Q_{i,w}^L(s_i, a_i, p_i) + a_i(r_i^L(\mathbf{s}, X) - p_i(s_i))$ and $Q_{i,w}^{IL}(s_i, a_i, p_i, r_i^L, X) = Q_{i,w}^L(s_i, a_i, p_i)$ for $s_i \neq \bar{s}_i$. Then the Iterative Linear-Whittle index is $w_i^{IL}(s_i, p_i, X) = \min_w \{w | Q_{i,w}^{IL}(s_i, 0, p_i, r_i^L, X) > Q_{i,w}^{IL}(s_i, 1, p_i, r_i^L, X)\}$, where $r_i^L(\mathbf{s}, X) = R_{\text{glob}}(\mathbf{s}, \mathbf{e}_i \vee \bigvee_{l=1}^j \mathbf{e}_{x_l}) - R_{\text{glob}}(\mathbf{s}, \bigvee_{l=1}^j \mathbf{e}_{x_l})$*

**Definition 5.** *Iterative Shapley-Whittle - Consider an instance of* RMAB-G $(\mathcal{S}, \mathcal{A}, R_i, R_{\text{glob}}, P_i, \gamma)$, *with $u_i(s_i)$ defined by Equation 3. Construct a new state, $\bar{s}_i$, with $P_{\bar{s}_i} = P_i$. Let $Q_{i,w}^{IS}(\bar{s}_i, a_i, u_i, r_i^S, X) = Q_{i,w}^S(s_i, a_i, u_i) + a_i(r_i^S(s_i, X) - u_i(s_i))$ and $Q_{i,w}^{IS}(s_i, a_i, u_i, r_i^S, X) = Q_{i,w}^S(s_i, a_i, u_i)$ for $s_i \neq \bar{s}_i$. Then the Iterative Shapley-Whittle Index is $w_i^{IS}(s_i, u_i, X) = \min_w \{w | Q_{i,w}^{IS}(s_i, 0, u_i, r_i^S, X) > Q_{i,w}^{IS}(s_i, 1, u_i, r_i^S, X)\}$. Here $r_i(s_i, X)$ is computed as:*

$$\min_{\mathbf{s}' | s_i' = s_i} \sum_{\substack{\mathbf{a} \in \{0,1\}^N, \|\mathbf{a}\|_1 \leq K-2 \\ a_i = a_{X_1} = a_{X_2} = \cdots = a_{X_j} = 0}} \frac{\|\mathbf{a}\|_1! (n - \|\mathbf{a}\|_1 - 1)!}{\frac{N!}{(N-K-1)!}} (R_{\text{glob}}(\mathbf{s}', \mathbf{a} \vee \mathbf{e}_i \vee X) - R_{\text{glob}}(\mathbf{s}', \mathbf{a} \vee X)) \quad (8)$$

By incorporating the arms pulled, $X$, both policies better estimate the present reward. Here, $Q_{i,w}^{IL}(\bar{s}_i, a_i, p_i, r_i^L, X)$ represents the expected discounted reward when receiving $r_i^L(\mathbf{s}, X)$ currently and $p_i(s_i)$ in the future (similarly for Shapley-Whittle). To model this, we construct a new state, $\bar{s}_i$, which mirrors $s_i$ except for the reward in the current timestep. The Iterative Linear-Whittle and Shapley-Whittle policies then iteratively select arms that maximize $w_i^{IL}$ and $w_i^{IS}$ respectively and stop after selecting $K$ arms.

## 5.3 MCTS-based Approaches

To improve upon the greedy selection done by iterative policies, we combine Monte Carlo Tree Search (MCTS) with Whittle indices. MCTS allows us to look beyond greedy selection by searching through various arm combinations. We propose two policies: MCTS Linear-Whittle and MCTS Shapley-Whittle. For both, we search through arm combinations, $\mathbf{a}$, compute the present value ($R(\mathbf{s}, \mathbf{a})$), and estimate the future value via Linear- and Shapley-Whittle indices. Nodes in the MCTS tree correspond to arm combinations, edges to individual arms, and leaf nodes to actions $\mathbf{a}$.

Our implementation of MCTS matches the upper confidence tree algorithm for selection, expansion, and backpropogation [20], while we differ for rollout (details in Appendix C). During rollout, we select nodes according to an $\epsilon$-greedy algorithm [21] and leverage the Linear- and Shapley-Whittle policies for greedy selection. We do so because $\epsilon$-greedy is a simple strategy that could potentially improve upon random rollouts. Once at a leaf node, the reward is $r = R(\mathbf{s}, \mathbf{a}) + \sum_{i=1}^N [Q_{i,0}^L(s_i, a_i) - p_i(s_i)a_i - R_i(s_i, a_i)]$ for MCTS Linear-Whittle and $r = R(\mathbf{s}, \mathbf{a}) + \sum_{i=1}^N [Q_{i,0}^S(s_i, a_i) - u_i(s_i)a_i - R_i(s_i, a_i)]$

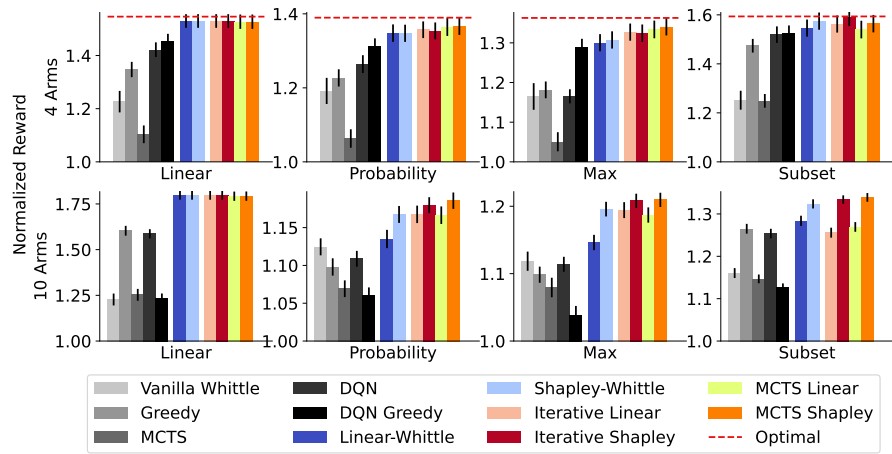

Figure 1: We compare baselines to our index-based and adaptive policies across four reward functions. All of our policies outperform baselines. Across all rewards, our best policy is within 3% of optimal for $N = 4$. Among our policies, Iterative and MCTS Shapley-Whittle consistently perform best.

for MCTS Shapley-Whittle. Here, $R(\mathbf{s}, \mathbf{a})$ is the known current value, and $\sum_{i=1}^N (Q_{i,0}^L(s_i, a_i) - p_i(s_i)a_i - R_i(s_i, a_i))$ is the estimated future value when accounting for the present reward $R(\mathbf{s}, \mathbf{a})$.

## 6 Experiments

We evaluate our policies across both synthetic and real-world datasets. We compare our six policies (Section 4.1, Section 5.2, and Section 5.3) to the following baselines (more details in Appendix B):

1. **Random** - We uniform randomly select $K$ arms at each timestep.
2. **Vanilla Whittle** - We run the vanilla Whittle index, which optimizes only for $R_i(s_i, a_i)$.
3. **Greedy** - We select actions which have the highest value of $p_i(s_i)$.
4. **MCTS** - We run MCTS up to some depth, $cK$, where $c$ is a constant.
5. **DQN** - We train a Deep Q Network (DQN) [22], and provide details in Appendix B.
6. **DQN Greedy** - Inspired by Theorem 3, we first learn a DQN then greedily optimize $Q(\mathbf{s}, \mathbf{a})$.
7. **Optimal** - We compute the optimal policy through value iteration when $N \leq 4$.

We run all experiments for 15 seeds and 5 trials per seed. We report the normalized reward, which is the accumulated discounted reward divided by that of the random policy. This normalization is due to differences in magnitudes within and across problem instances and we use discounted rewards due to prior work [11]. We additionally compute the standard error across all runs.

### 6.1 Results with Synthetic Data

To understand performance across synthetic scenarios, we develop synthetic problem instances that leverage each of the following global reward functions (details on $m_i$ and $Y_i$ choices in Appendix A):

1. **Linear** - Given $m_1, \cdots, m_N \in \mathbb{R}$, then $R_{\text{glob}}(\mathbf{s}, \mathbf{a}) = \sum_{i=1}^N m_i s_i a_i$.
2. **Probability** - Given $m_1, \cdots, m_N \in [0, 1]$, then $R_{\text{glob}}(\mathbf{s}, \mathbf{a}) = 1 - \prod_{i=1}^N (1 - m_i a_i s_i)$.
3. **Max** - Given $m_1, \cdots, m_N \in \mathbb{R}$, then $R_{\text{glob}}(\mathbf{s}, \mathbf{a}) = \max_i s_i a_i m_i$.
4. **Subset** - Given sets, $Y_1, \cdots, Y_N \subset \{1, \cdots, m\}, m \in \mathbb{Z}$, then $R_{\text{glob}}(\mathbf{s}, \mathbf{a}) = |\cup_{i|s_i a_i = 1} Y_i|$.

We select these rewards, as they range from linear or near-linear (*Linear* and *Probability*) to highly non-linear (*Max* and *Subset*). The reward functions can also be employed to model real-world situations; the *Probability* reward models food rescue and the *Subset* reward models peer review (see Section 3). For all rewards, we let $R_i(\mathbf{s}, \mathbf{a}) = s_i/N$ and $\mathcal{S} = \{0, 1\}$, which matches prior work [12]. We construct synthetic transitions parameterized by $q$, so $P_i(0, 0, 1) \sim U(0, q)$, where $U$ is the uniform distribution (see Appendix A). We construct transition probabilities so that having $a_i = 1$ or

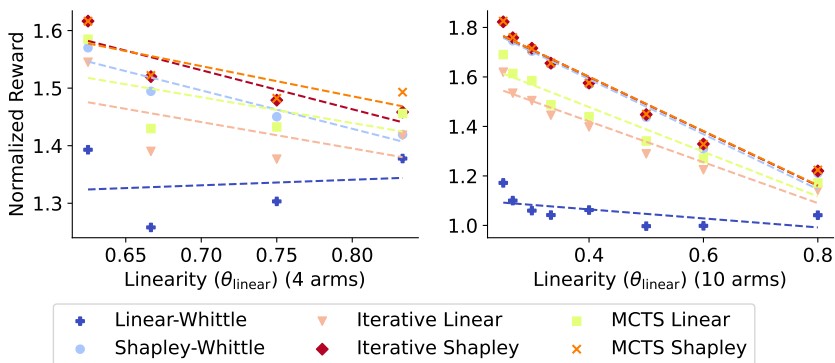

Figure 2: We plot policy performance for instances of the *Subset* reward which vary in linearity. We see that the Iterative and MCTS Shapley-Whittle policies outperform alternatives for non-linear rewards (small $\theta_{\text{linear}}$) while policy performances converge for linear rewards (large $\theta_{\text{linear}}$).

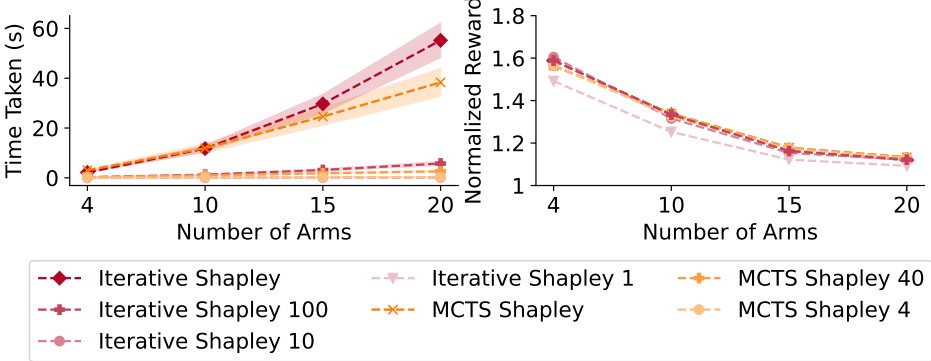

Figure 3: We compare the efficiency and performance of Iterative Shapley-Whittle methods which vary in Monte Carlo samples used for Shapley estimation and MCTS Shapley-Whittle methods which vary in MCTS iterations. While Iterative Shapley-Whittle is the slowest, decreasing the number of Monte Carlo samples can improve efficiency without impacting performance.

$s_i = 1$ only improves transition probabilities (see Wang et al. [11]). We vary $q$ in Appendix G and find that our choice of $q$ does not impact findings. Unless otherwise specified, we fix $K = N/2$.

Figure 1 shows that our policies consistently outperform baselines. Among our policies, Iterative and MCTS Shapley-Whittle policies perform best. For $N = 10$, Iterative and MCTS Shapley-Whittle significantly outperform baselines (paired t-test $p < 0.04$) and index-based policies except for the *Linear* reward ($p < 0.001$). DQN-based policies regress from on average $4\%$ worse for $N = 4$ to on average $9\%$ worse for $N = 10$ compared to our best policy, which occurs due to the exponential state space. In Appendix I, we consider more combinations of $N$ and $K$ and find similar results.

To understand policy performance across rewards, we analyze how reward linearity impacts performance. We quantify linearity via $\theta_{\text{linear}}$ (see Theorem 6) and compute $\theta_{\text{linear}}$ for *Subset* instances varying in $Y_i$ (see Appendix A). For non-linear reward functions (small $\theta_{\text{linear}}$), Iterative and MCTS-Shapley-Whittle are best, outperforming Linear-Whittle by $56\%$, while for near-linear reward functions (large $\theta_{\text{linear}}$), all policies are within $18\%$ of each other (Figure 2, $N = 10$). Appendix H shows that, for some rewards, adaptive policies outperform non-adaptive policies by $50\%$.

To understand the tradeoff between efficiency and performance we plot the time taken and normalized reward for various $N$ (Figure 3). We evaluate variants of Iterative Shapley-Whittle and MCTS Shapley-Whittle because both perform well yet are computationally expensive. For Iterative Shapley-Whittle, 1000 samples are used for Shapley estimation by default, so we consider variants that use 100, 10, and 1 samples. For MCTS Shapley-Whittle, we run 400 iterations of MCTS by default, so we consider variants that run 40 and 4 iterations. Iterative Shapley-Whittle policies are the slowest

|  | Vanilla Whittle | Greedy | MCTS | DQN | Linear Whittle |
|---|---|---|---|---|---|
| Notifications | $0.975 \pm 0.012$ | $1.829 \pm 0.016$ | $1.047 \pm 0.012$ | $1.547 \pm 0.025$ | $\mathbf{1.932 \pm 0.017}$ |
| Phone Calls | $0.989 \pm 0.009$ | $1.228 \pm 0.008$ | $1.070 \pm 0.016$ | $1.209 \pm 0.007$ | $1.288 \pm 0.008$ |

|  | Shapley Whittle | Iter. Linear | Iter. Shapley | MCTS Linear | MCTS Shapley |
|---|---|---|---|---|---|
| Notifications | $1.931 \pm 0.016$ | $1.921 \pm 0.017$ |  |  |  |
| Phone Calls | $1.290 \pm 0.008$ | $1.287 \pm 0.009$ | $\mathbf{1.294 \pm 0.009}$ | $1.291 \pm 0.008$ | $1.293 \pm 0.008$ |

Table 1: We evaluate policies in real-world food rescue contexts by developing situations that mirror notifications ($N = 100, K = 50$) and phone calls ($N = 20, K = 10$). Our policies outperform baselines for both situations and achieve similar results due to the nearly linear reward function.

but can be made faster without impacting performance by reducing the number of samples. Our results demonstrate how we can develop efficient adaptive policies which perform well.

## 6.2 Results with Real-World Data in Food Rescue

To evaluate our policies in real-world contexts, we apply our policies to a food rescue dataset. We leverage data from a partnering multi-city food rescue organization and construct an RMAB-G instance using their data (details in Appendix D). Here, arms correspond to volunteers, and states correspond to engagement, with $s_i = 1$ indicating an engaged volunteer. Notifications are budget-limited because sending out too many notifications can lead to burned-out volunteers.

We compute transition probabilities from volunteer participation data and compute match probabilities from volunteer response rates. Using this, we study whether our policies maintain a high trip completion rate and volunteer engagement. We assume that food rescue organizations are indifferent to the tradeoff between trip completion and engagement. Because such an assumption might not hold in reality, we explore alternative tradeoffs in Appendix I. If volunteer $i$ is both notified ($a_i = 1$) and engaged ($s_i = 1$), then they match to a trip with probability $m_i$. We do so because unengaged volunteers are less likely to accept food rescue trips. Under this formulation, our global reward is the probability that any volunteer accepts a trip, which is exactly the *Probability* reward. For engagement, we let $R_i(s_i, a_i) = s_i/N$, which captures the fraction of engaged volunteers. We model two food rescue scenarios, and detail scenarios with more volunteers in Appendix J:

**Notifications** - Food rescue organizations deliver notifications to volunteers as new trips arise [7]. To model this scenario, we consider $N = 100$ volunteers and $K = 50$ notifications.

**Phone Calls** - When a trip remains unclaimed after notifications, food rescue organizations call a small set of experienced volunteers. We model this with $N = 20, K = 10$, and using only arms corresponding to experienced volunteers with more than 100 trips completed (details in Appendix D).

In Table 1, we compare our policies against baselines and find that our policies achieve higher normalized rewards. All of our policies outperform all baselines by at least 5%. Comparing between policies, we find that Iterative and MCTS Shapley-Whittle perform best, performing slightly better than the Shapley-Whittle policy. Currently, food rescue organizations perform notifications greedily, so an RMAB-G framework could improve the engagement and trip completion rate.

## 7 Related Works and Problem Setup

**Multi-armed Bandits** In our work, we explore multi-armed bandits with two properties: combinatorial actions [23] and non-separable rewards. Combinatorial bandits can be solved through follow-the-perturbed-leader algorithms [24], which are necessary because the large action space causes traditional linear bandit algorithms to be inefficient [21]. Prior work has tackled non-separable rewards in slate bandits [25], through an explore-then-commit framework, and adversarial combinatorial bandits [26], from a minimax optimality perspective. In this work, we leverage the structure of RMAB-Gs to tackle both combinatorial actions and non-separable rewards.

**Submodular Bandits** Additionally related to our work are prior investigations into submodular bandits. Submodular bandits combine submodular functions with bandits and have been investigated from an explore-then-commit perspective [27] and through upper confidence bound (UCB) algorithms [28]. Similar to prior work, we focus on optimizing submodular functions under uncertainty. However, prior work focuses on learning the reward structure when the submodular function is unknown, while we focus on learning optimal decisions when state transitions are stochastic.

**Restless Multi-Armed Bandits** RMABs extend multi-armed bandits to situations where unpulled arms can transition between states [1]. While finding optimal solutions to this problem is PSPACE-Hard [29], index-based solutions are efficient and asymptotically optimal [2]. As a result, RMABs have been used across a variety of domains [4, 3, 30] and have been deployed in public health scenarios [12]. While prior work has relaxed assumptions by considering RMABS that have contexts [31], combinatorial arms [32], graph-based interactions [33], and global state [34], we focus on combinatorial situations with a global reward.

## 8    Limitations and Conclusion

**Limitations** - While we present approximation bounds for index-based policies, our adaptive policies lack theoretical guarantees. Developing bounds for adaptive policies is difficult due to the difficulty in leveraging Whittle index optimality. However, these bounds may explain the empirical performance of adaptive policies. Our empirical analysis is limited to one real-world dataset (food rescue). We aimed to mitigate this by exploring different scenarios within food rescue, but empirical evaluation in domains such as peer review would be valuable. Finally, evaluating policies when global rewards are neither submodular nor monotonic would improve our understanding of RMAB-G.

**Conclusion** - RMABs fail to account for situations with non-separable global rewards, so we propose the RMAB-G problem. We tackle RMAB-G by developing index-based, iterative, and MCTS policies. We prove performance bounds for index-based policies and empirically find that iterative and MCTS policies perform well on synthetic and real-world food rescue datasets. Our results show how RMABs can extend to scenarios where rewards are global and non-separable.

## Acknowledgements

We thank Gati Aher and Gaurav Ghosal for the comments they provided on an earlier draft of this paper. We additionally thank Milind Tambe for his comments. This work was supported in part by NSF grant IIS-2200410. Co-author Raman is also supported by an NSF GRFP Fellowship. Co-author Shi is supported in part by a Pitt Mascaro Center for Sustainable Innovation fellowship.

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

## Broader Impact

In this paper, we analyze the use of restless multi-armed bandits to tackle general reward functions and apply this to real-world problems in food rescue. Our goal is to develop practically motivated and theoretically insightful algorithms. With this goal, we evaluate our algorithms on real-world data to understand whether such decision-making algorithms have real-world validity. Our proposed algorithms and analysis are a step towards improving the deployment of restless multi-armed bandit algorithms in real-world situations such as food rescue.

## A   Experimental Details

We provide additional details on our experiments and experimental conditions. For all experiments in the main paper, we run experiments for 15 seeds, and for each seed, we run experiments across 5 choices of initial state (5 trials). For all ablations in the Appendix, we run for 9 seeds and run experiments across 5 choices of initial state. Random seeds change the reward parameters ($m_i$ or $Y_i$) and the transitions. For all plots, we plot the normalized reward averaged across seeds along with the standard error. We compute the discounted reward with an infinite time horizon and normalize this as the improvement over the random policy. We do this so that we can standardize the magnitude of rewards, as such a magnitude can vary across rewards and selections of reward parameters. For all experiments $\gamma = 0.9$. We estimate the infinite-horizon discounted reward by computing the discounted reward for $T = 50$ timesteps; we do this because for $t > 50$, $\gamma^t \leq 0.005$, and so we can compute the discounted reward with only a small difference from the infinite time horizon.

To construct synthetic transitions, we consider a parameter $q$, and let $P_i(0,0,1) \sim U(0,q)$ where $U$ is the uniform distribution. Then $P_i(1,0,1) \sim U(P_i(0,0,1),1)$ and $P_i(0,1,1) \sim U(P_i(0,0,1),1)$. Finally, $P_i(1,1,1) \sim U(\max(P_i(1,0,1), P_i(0,1,1)),1)$. We construct transitions in such a way so that being in $s_i = 1$ or playing action $a_i = 1$ can only help with these synthetic transitions, which matches from prior work [11]. We vary $q$ in Appendix G, and find similar results across selections of $q$.

We additionally detail some of the choices for parameters for each of the rewards used in Section 6.1. For the *Linear*, *Probability*, and *Max* rewards, we select $m_1, \cdots, m_N \sim U(0,1)$. For the *Probability* distribution, such a selection is natural (as probabilities are between 0 and 1), while for the *Linear* and *Max* reward, we find similar results no matter the distribution of $m_i$ chosen. We consider different selections for the sizes of the subset and use these different sizes to investigate the impact of linearity on performance. By default, we let $m = 20$ and let $Y_i$ be sets of size 6. We select such a size so that $R_{\mathrm{glob}}(\mathbf{s}, \mathbf{a})$ is non-linear in $|Y_i|$, as when $|Y_i|$ is too small the problem becomes more linear (because $Y_i \cap Y_j$ is empty).

To understand the impact of linearity, we construct the sets $Y_i$ in such a way to control $\theta_{\mathrm{linear}}$. In particular, we consider two parameters $a$ and $b$. We construct $K$ (where $K$ is the budget) sets which where $Y_i = \{1, 2, \cdots, b\}$, and we construct $N - K$ disjoint sets which are size $a < b$. For the $N - K$ disjoint sets, they would be $\{1, 2, \cdots, a\}, \{a + 1, a + 2 \cdots, 2a\}, \cdots$. Under this construction, $\theta_{\mathrm{linear}} = \frac{b}{Ka}$. We then vary $b \in \{3, 4, 5\}$, and vary $a \in \{1, \cdots, b - 1\}$, then compute the performance of each policy for each of these rewards. Constructing sets in this way allows us to directly measure the impact of linearity on performance, as we can control the linearity of the reward function by varying $a$ and $b$.

We run all experiments on a TITAN Xp with 12 GB of GPU RAM running on Ubuntu 20.04 with 64 GB of RAM. Each policy runs in under 30 minutes, and we use 300-400 hours of computation time for all experiments. We develop DQN policies using PyTorch [35].

## B   Baseline Details

We provide additional details on select baselines here:

1. **DQN** - We develop a Deep Q Network (DQN), where a state consists of the set of arms pulled and the current state of all arms. The arms correspond to a new arm to pull; we structure the DQN to avoid the number of actions being exponential in the budget (which is often linear in the number of arms). We train the DQN for 100 epochs and evaluate with

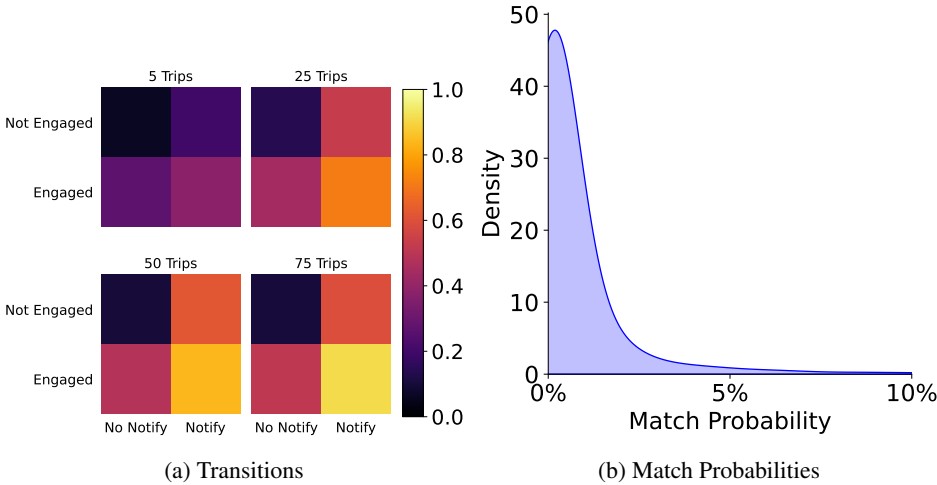

(a) Transitions           (b) Match Probabilities

Figure 4: We plot the (a) transition probabilities obtained by clustering volunteers with various levels of experience and (b) the distribution of match probabilities across all volunteers. We see that as volunteers gain more experience, they are more likely to become engaged when notified. However, we see that most volunteers have a low probability of responding, which motivates the need for large budgets in notification.

    other choices in Appendix F. We let the learning rate be $5 * 10^{-4}$, a batch size of 16, and an Adam optimizer. We use an MLP with a 128-dimension as our hidden layer, and use 2 such hidden layers. We use ReLu activation for all layers. We provide more details on training DQNs in our supplemental codebase.

2. **DQN Greedy** - Inspired by Theorem 3, we combine a DQN with greedy search. We first learn a DQN network where states directly correspond to the states of the arms, and actions correspond to subsets of arms pulled. Due to the large number of actions, such a method does not scale when the number of arms is increased (as the number of actions is exponential in the budget). Using this learned Q function, we greedily select arms one-by-one, until $K$ total arms are selected. We let the learning rate be $5 * 10^{-4}$, a batch size of 16, and an Adam optimizer.

3. **MCTS** - We run our MCTS baseline algorithm and use the upper confidence tree algorithm [20]. We run this for 400 iterations, and recurse until a depth of $2 * K$ is reached; we detail other choices for this in Appendix E. We select a depth of $2K$ so that arm selections are non-myopic. By letting the depth be $K$, the MCTS algorithm would simplify into a better version of greedy search, whereas we want to see if we can non-myopically select arms using MCTS.

## C   Policy Details

We provide details for each of our policies below. For Shapley-based methods, we approximate the Shapley value by computing 1000 random samples of combinations. We do this because computing the exact Shapley value takes exponential time in the number of arms. In Section 6, we consider the impact of other choices on the efficiency and performance of models.

We additionally provide the pseudo-code for our iterative algorithms (Algorithm 2) and our MCTS Shapley-Whittle and MCTS Linear-Whittle algorithms (Algorithm 1). For our MCTS-based policies, we note that this follows from the typical upper confidence trees framework, varying only in the rollout step. We let $\epsilon = 0.1$ for the rollout function, and let $c = 5$.

## D   Food Rescue Details

To apply the food rescue scenario to RMABs, we compute aggregate transition probabilities for volunteers through clustering. Each volunteer in the food rescue performs a certain number of trips,

---

**Algorithm 1** Monte Carlo Tree Search for RMAB-Gs

---

**Input:** State $\mathbf{s}$, reward function $R(\mathbf{s}, \mathbf{a})$, Q functions $Q^L_{i,w}(s_i, a_i, p_i)$ or $Q^S_{i,w}(s_i, a_i, u_i)$, and transitions $P_1, \cdots, P_N$, number of MCTS trials $M$, and hyperparameter $c$
**Output:** Action $\mathbf{a}$
Initialize a tree, $T$, with leafs representing $K$ arms pulled
Let $v_m$ represent the total value of a node $m$
Let $n_m$ represent the number of visits to a node $m$
**for** $t = 1$ **to** $M$ **do**
    Let $m$ be the root of $T$
    Let $l$ be empty set
    **while** $m$ is non-leaf **do**
        Add $m$ to $l$
        **if** $m$ has an unexplored child **then**
            **while** $m$ is non-leaf **do**
                With probability $\epsilon$, let $m'$ be a random unexplored child of $m$
                Otherwise, let $m'$ be the unexplored child of $m$, with the largest value of $w_i(s_i)$, where $w_i(s_i)$ is either the Linear or Shapley-Whittle index
                Add $m'$ to $l$
                Let $m = m'$
            **end while**
        **else**
            Let $m' = \text{argmax}_{m'} \frac{v_{m'}}{n_{m'}} + c\sqrt{\frac{n_m}{n_{m'}}}$
            Let $m = m'$
        **end if**
    **end while**
    Compute action $\mathbf{a}$ from nodes played, $l$
    For MCTS Linear-Whittle, let $r = R(\mathbf{s}, \mathbf{a}) + \sum_{i=1}^{N}[Q^L_{i,w}(s_i, a_i, p_i) - p_i(s_i)a_i - R_i(s_i, a_i)]$
    For MCTS Shapley-Whittle, $r = R(\mathbf{s}, \mathbf{a}) + \sum_{i=1}^{N}[Q^S_i(s_{i,w}, a_i, u_i) - u_i(s_i)a_i - R_i(s_i, a_i)]$
    **for** $m \in l$ **do**
        Update $w_m \leftarrow w_m + r$
        Update $n_m \leftarrow n_m + 1$
    **end for**
**end for**
**return** The action corresponding to the leaf, $m$, with the highest $\frac{w_m}{n_m}$

---

and we use this information to construct clusters. We ignore volunteers who have performed 1 or 2 trips, as those volunteers have too little information to construct transition functions for. We construct 100 clusters of roughly equal size, where each cluster consists of volunteers who perform a certain number of trips; for example, the 1st cluster consists of all volunteers who perform 3 trips, and the 90th cluster is those who perform 107-115 trips. From this, we compute transition probabilities by letting the state (which represents engagement) denote whether a volunteer has completed a trip in the past two weeks. We then see if completing a trip in the current two weeks impacts the chance that they complete a trip in the next two weeks. We use this as a proxy for engagement due to the lack of ground truth information on whether a volunteer is truly engaged. We average this over all volunteers in a cluster and across time periods to compute transition probabilities. We present examples of such transition probabilities in Figure 4. To compute the corresponding match probabilities for each volunteer, we compute the fraction of notifications they accept, across all notifications seen. This way, each arm corresponds to a cluster of volunteers, with a transition probability and match probability. We compute match probabilities on a per-volunteer basis so that multiple arms can have the same transition probability (corresponding to volunteers being in the same cluster) while differing in match probability. Such a situation corresponds to the real world as well; not all volunteers who complete between 107-115 trips have the same match probability.

We analyze the distribution of match probabilities for each volunteer, across all trips that they are notified for. We find that most volunteers accept less than 2% of the trips they've seen, while a few volunteers answer more than 10% (Figure 4). We note the difference between our two scenarios:

**Algorithm 2** Iterative Policies for RMAB-Gs

---

**Input:** State $\mathbf{s}$, reward function $R(\mathbf{s}, \mathbf{a})$, Whittle functions $w_i^L(s_i, p_i, X)$ or $w_i^S(s_i, u_i, X)$, and transitions $P_1, \cdots, P_N$
**Output:** Action $\mathbf{a}$
Let $X = \emptyset$
**for** $t = 1$ **to** $K$ **do**
   Let $v_i = 0$ for $1 \le i \le N$
   **for** $i = 1$ **to** $N$ **do**
      **if** $i \notin X$ **then**
         Update $v_i = w_i^L(s_i, p_i, X)$ or $v_i = w_i^S(s_i, u_i, X)$
      **end if**
   **end for**
   **if** $\max v > 0$ **then**
      Update $X \leftarrow X \cup \{\arg \max v\}$
   **end if**
**end for**
Let $\mathbf{a} \in \{0, 1\}^N$, with $a_i = 1 \forall i \in S$

---

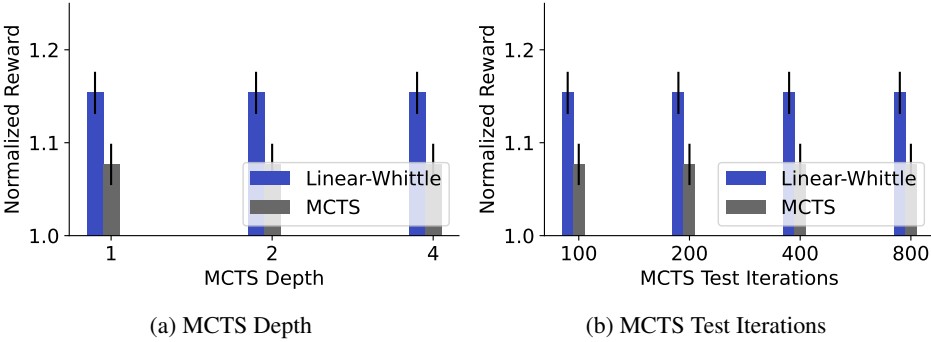

(a) MCTS Depth          (b) MCTS Test Iterations

Figure 5: We compare the performance of vanilla MCTS algorithms when varying (a) depth of exploration and (b) the number of test iterations. We find that, no matter the choice of depth or test iterations, MCTS performs worse than Linear-Whittle. Moreover, as expected, increasing test iterations improves performance, while lower depth improves MCTS performance due to the difficulty in estimating rewards from transitions.

notifications and phone calls. For notifications, we consider all volunteers and notify a large fraction of these volunteers, while for phone calls, we consider only volunteers who have completed more than 100 trips, as they generally have a higher match probability. We then cluster those volunteers into 20 groups and similarly compute transition probabilities. We do this because food rescue organizations target more active volunteers when making phone calls due to the limited number of phone calls that can be made.

## E   Vanilla MCTS Experiments

To understand the impact of our choice of hyperparameters for the MCTS baseline algorithm, we vary the depth of the MCTS search, along with the number of iterations. We vary the depth of MCTS in $\{1K, 2K, 4K\}$ (while setting the iterations to $400$), and we vary the number of iterations run for MCTS in $\{50, 100, 200, 400\}$ (while searching up to depth $2K$). For all runs, we let $K = 5$ and $N = 10$ for the *Max* reward.

In Figure 5, we see that increasing MCTS depth only lowers performance, while increasing the number of iterations MCTS is run for increases performance. This is due to the stochasticity in rewards as the depth of exploration increases; there are $|\mathcal{S}|^N$ possible combinations of states, making exploration difficult. We find that for any combination of iterations and depth, the performance of MCTS lags significantly behind Linear-Whittle. While running MCTS for additional iterations

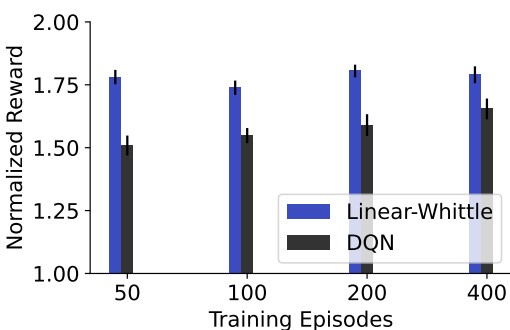

Figure 6: We compare the impact of additional training epochs on the performance of a DQN baseline. We find that, regardless of the number of training episodes, Linear-Whittle outperforms the DQN baseline. Additional training episodes do not result in a significantly better performance (compared with Linear-Whittle), showing that additional training episodes do not lead to a much smaller gap between Linear-Whittle and the DQN baseline.

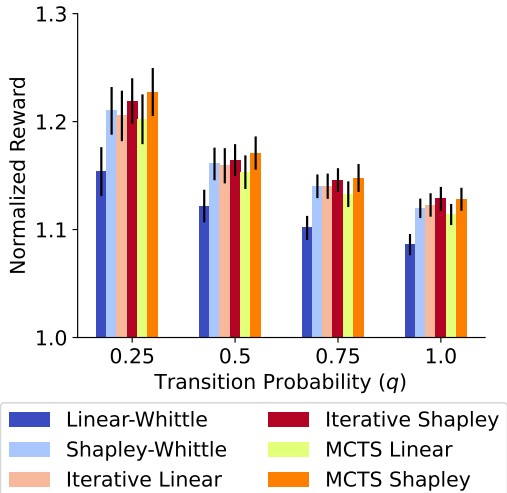

Figure 7: We assess the impact of changing $q$, which parametrizes transition probabilities. Smaller $q$ makes it less likely that arms transition to the 1 state. We find that, regardless of the transition probability chosen, MCTS Shapley-Whittle policies improve upon all other policies.

would be helpful, this comes at a cost of time, so we run MCTS for $400$ iterations throughout our experiments to balance these.

# F   Deep Q Networks

We compare the performance of our reinforcement learning baseline (DQN) to the Linear-Whittle policy as we vary the number of epochs that the DQN is trained for. We vary the number of train epochs in $\{50, 100, 200, 400\}$ and plot our results in Figure 6. We see that increasing the number of iterations does not provide a big advantage in performance; this is because DQN fails to learn past $50$ epochs, and so additional training steps do not lead to much better performance. Moreover, we see that, no matter the selection of train iterations, DQN performs worse than Linear-Whittle. We believe the reason for this is the stochasticity in the dataset; because the number of actions is large, it becomes difficult to learn state values in this scenario, which leads to the poor performance of DQN.

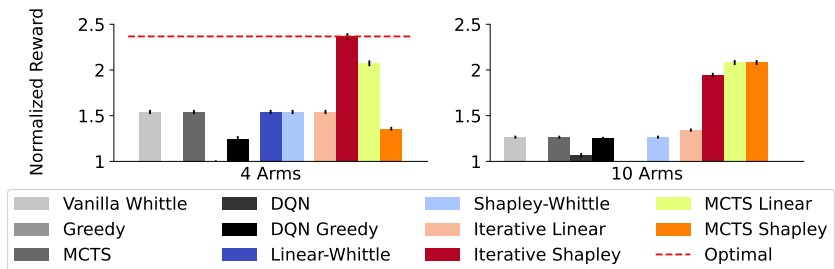

Figure 8: We construct situations where adaptive policies are needed to perform better than baselines. When $N = 4$, Iterative Shapley-Whittle significantly outperforms alternatives, and when $N = 10$, MCTS Shapley-Whittle outperforms alternatives.

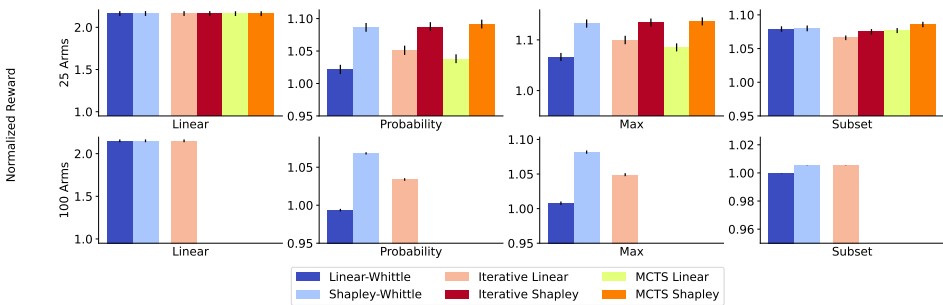

Figure 9: We repeat Figure 1 for $N = 25$ and $N = 100$. We find that many of the same trends appear here; namely, that MCTS Shapley-Whittle outperforms all other policies, and Shapley-based policies do better in general.

## G  Synthetic Transitions

Throughout our experiments, we sample synthetic transitions, with a maximum probability for $P_i(0, 1, 0)$ of $q$; we vary the value of $q$ to understand if this impacts the performance of policies. We vary $q \in \{0, 0.25, 0.5, 0.75, 1.0\}$, and evaluate the performance of our policies on the *Max* reward.

We find that, across all values of $q$, the MCTS Shapley-Whittle policy performs best. We find large gaps between MCTS Shapley-Whittle and Linear-Whittle for small $q$; in these scenarios, the transitions are more stochastic, leading to greater impact from MCTS Shapley-Whittle. However, regardless of the choice of $q$, our conclusion remains that Iterative and MCTS Shapley-Whittle policies outperform other alternatives.

## H  Other Constructions of Transitions

We investigate whether situations detailed in Lemma 8 can be tackled by adaptive algorithms. We construct a situation for the *Max* reward where transitions are $P_i(s, 1, 0) = 1$ and $P_i(s, 0, s) = 1$; pulling an arm always leads to $s_i = 0$, and otherwise, arms stay in their current state. We design such a situation to see whether adaptive policies can perform well in situations where index-based policies are proven to perform poorly. We let the values $m_1 = m_2 = 1$, then let the other choices of $m_i = 0$. We do this to encourage arms to be pulled one at a time, rather than pulling $m_1$ and $m_2$ together. In Figure 8, we compare the performance of policies in this situation and see that Iterative Shapley-Whittle significantly outperforms all other policies when $N = 4$, and that MCTS Linear- and Shapley-Whittle outperform all other policies when $N = 10$. When $N = 10$, we see that all non-adaptive Whittle policies achieve similar performance, demonstrating the need for adaptive policies.

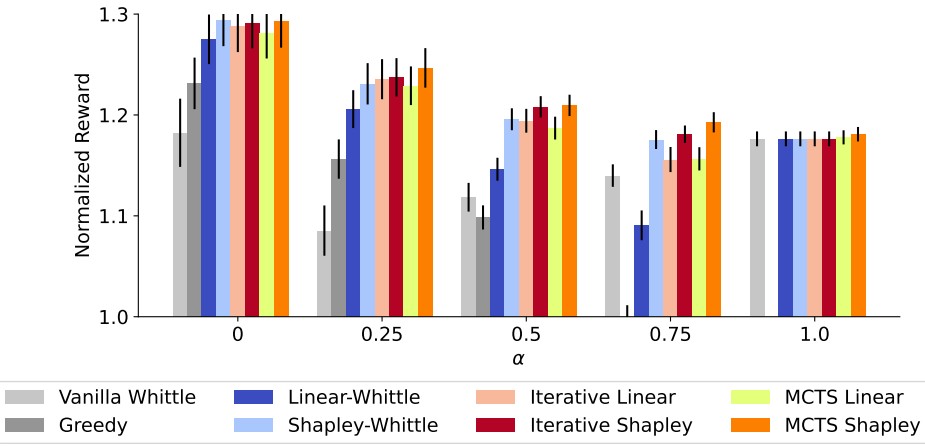

Figure 10: We vary the parameter $\alpha$, which modulates between the individual rewards $R_i$ and the global reward $R_{\mathrm{glob}}$. When $\alpha = 1$, all policies perform similarly, while for $\alpha \in \{0.25, 0.5, 0.75\}$, we see that Linear-Whittle performs worse than other policies. Across values of $\alpha$, baselines perform worse than all policies, which is most notable when $\alpha = 0$.

## I  Other Combinations of Arms and Budget

**More arms**  We evaluate the performance of policies when $N = 25$ or $N = 100$ and plot this in Figure 9. We run Iterative Shapley-Whittle, MCTS Linear-Whittle, and MCTS Shapley-Whittle policies for $N = 25$ and refrain from running them when $N = 100$ due to the time needed. We find that MCTS Shapley-Whittle policies perform better than alternatives, and that they perform significantly better than Linear-Whittle policies. The results with $N = 25$ and $N = 100$ confirm the trends from Section 6.1 even when the number of arms is large.

**Weighting Individual and Global Rewards**  Throughout our experiments, we weight the impact of the global reward, $R_{\mathrm{glob}(\mathbf{s},\mathbf{a})}$, and the individual reward, $R_i(s_i, a_i)$, equally. We evaluate the impact of various weighting factors so that $R(\mathbf{s}, \mathbf{a}) = (1 - \alpha)R_{\mathrm{glob}}(\mathbf{s}, \mathbf{a}) + \alpha \sum_{i=1}^{N} R_i(\mathbf{s}, \mathbf{a})$. By default, $\alpha = 0.5$ throughout our experiments, so we vary $\alpha \in \{0, 0.25, 0.5, 0.75, 1.0\}$ and evaluate the reward for baselines and our policies on the maximum reward.

In Figure 10, we see that, regardless of the choice of $\alpha$, our policies outperform baselines. We additionally see that the Shapley-Whittle and Iterative Shapley-Whittle perform well across the choices of $\alpha$, implying that the results we found with $\alpha = 0.5$ are not limited to one choice of $\alpha$.

**Varying Budget**  Throughout our experiments, we let the budget, $K = 0.5N$, so we analyze the impact of varying $K$ when fixing $N = 10$. We vary $K \in \{2, 5, 8\}$, and note that with $K = 10$ or $K = 0$, all policies perform the same (as all or no arms are selected). We see that, across all choices of budget, Linear-Whittle performs worse than all other policies (Figure 11). Moreover, we see again that regardless of the choice of $K$, MCTS Shapley-Whittle is the best policy.

## J  Food Rescue More Arms

In addition to the experiments in Section 6.2, we evaluate the impact of increasing the number of arms in food rescue on the performance of various policies. We consider a situation with $N = 1000$ and $K = 250$, for the notification scenario, and another situation with $N = 250$ and $K = 5$, which corresponds to the phone calls scenario. We plot both situations in Figure 12, and we see that all policies perform similarly in both of these situations. When the number of arms is this large, all policies will select the top arms as arms are likely to have a large match probability due to the sheer number of arms. We can differentiate between policies in the notification scenario, where Shapley-Whittle policies perform better than Linear-Whittle policies, but in the phone calls settings, all policies perform similarly. Additionally, due to the large budget, the match probability will be

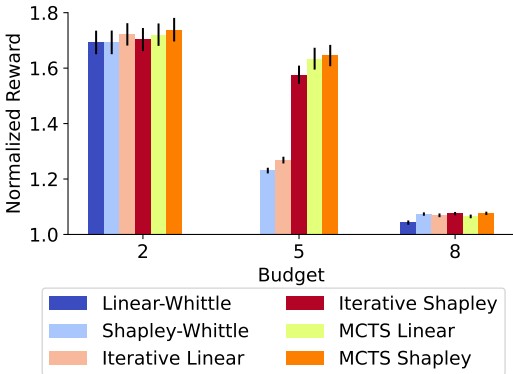

Figure 11: We vary the budget, $K$, while fixing $N = 10$ for the maximum reward. We see that, for any choice of $K$, all policies have a larger reward than greedy, and that Linear-Whittle policies perform worse than other policies. Moreover, for $K = 5$ and $K = 8$, Iterative Shapley-Whittle policies outperform Iterative Linear-Whittle policies, while for $K = 2$ policies, Iterative Linear-Whittle policies perform well, potentially due to the linearity of the problem with small $K$.

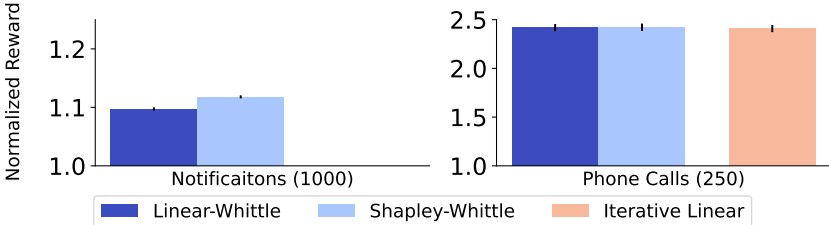

Figure 12: We compare the performance of policies on the food rescue dataset for two situations: (a) notifications, when $N = 1000$ and $K = 500$, and (b) phone calls, when $N = 250$ and $K = 50$. We find that when $N$ is large, all policies perform similarly, due to the linearity of the problem for large $N$.

large, so policies should focus on engagement in this scenario. In reality, the choice of whether to use pre-computed index policies or adaptive policies depends on the time available, the size of the problem, and the structure of the reward.

## K    Proofs of Problem Difficulty

**Theorem 1.** *The restless multi-armed bandit with global rewards is PSPACE-Hard.*

*Proof.* We reduce RMAB to RMAB-G. Note that RMAB-G aims to maximize

$$\sum_{t=0}^{\infty} \gamma^t (R_{\text{glob}}(\mathbf{s}, \mathbf{a}) + \sum_{i=1}^{N} R_i(s_i, a_i)) \tag{9}$$

Consider an RMAB instance with reward $R_i'(s_i, a_i)$. We can construct an instance of RMAB-G with $R_i = R_i'$ and with $R_{\text{glob}}(\mathbf{s}, \mathbf{a}) = 0$. Therefore, we have reduced RMAB to RMAB-G. Because RMAB is PSPACE-Hard [29], RMAB-G is as well. $\square$

**Theorem 2.** *No polynomial time algorithm can achieve better than a $1 - \frac{1}{e}$ approximation to the restless multi-armed bandit with global rewards problem.*

*Proof.* Consider any submodular function $F(S) : 2^\Omega \to \mathbb{R}$ and a cardinality constraint $|S| \le K$. Unless $P = NP$, maximizing $F(S)$ with a cardinality constraint can only be approximated up to a factor of $1 - \frac{1}{e}$ [36]. Next, we will show that we can reduce submodular maximization to RMAB-G.

For a given submodular function $F$, we construct an instance of RMAB-G with $\mathcal{S} = \{0, 1\}$. Let $\mathbf{s}^{(0)} = \mathbf{1}$, where $\mathbf{1}$ is the vector of $N$ 1s. Finally let $P_i(s, a, 0) = 1$, so that all arms are in state 0 for timesteps $t > 0$. Let $R(\mathbf{s}, \mathbf{a}) = 0$ if $\mathbf{s} \neq \mathbf{1}$, and let $R(\mathbf{s}, \mathbf{a}) = F(\{i | a_i = 1\})$ otherwise. Note that only at time $t = 0$ can you achieve any reward, as afterwards, $F(S) = 0$.

The optimal solution to the RMAB-G problem is the action, $\mathbf{a}$ that maximizes $F(\{i | a_i = 1\})$ subject to the constraint, $\sum_{i=1}^{N} a_i \leq K$. Note that this is the same as the cardinality constraint for submodular maximization, $|S| = \sum_{i=1}^{N} a_i$, along with the function of interest, $F(S)$. Therefore, if we can efficiently solve RMAB-G, then we can efficiently solve the original submodular maximization problem in $F$. Because we cannot do better than a $1 - \frac{1}{e}$ approximation to the submodular maximization problem in polynomial time, we cannot do better than a $1 - \frac{1}{e}$ approximation to the RMAB-G problem. $\square$

**Lemma 9.** *Let $\mathcal{P}$ be a set of transition functions, $P_1, \cdots, P_N \in \mathcal{S} \times \mathcal{A} \times \mathcal{S} \to [0, 1]$, where $N$ is the number of arms. Let $\mathcal{S} = \mathcal{A} = \{0, 1\}$. Let $Q_\pi(\mathbf{s}, \mathbf{a}) = \mathbb{E}_{(\mathbf{s}', \mathbf{a}') \sim (\mathcal{P}, \pi)}[\sum_{t=0}^{\infty} \gamma^t R(\mathbf{s}', \mathbf{a}'))] | \mathbf{s_0} = \mathbf{s}, \mathbf{a_0} = \mathbf{a}]$ for some policy $\pi : \mathcal{S} \to \mathcal{A}$, and let $\pi^*$ be the optimal policy. Let $V_{\pi^*}(\mathbf{s}) = \max_{\mathbf{a}} Q_\pi(\mathbf{s}, \mathbf{a})$. If $R(\mathbf{s}, \mathbf{a})$ is monotonic in $\mathbf{s}$ and $P_i(1, a, 1) \geq P_i(0, a, 1)$ for any $a$, then $V_{\pi^*}(\mathbf{s})$ is monotonic in $\mathbf{s}$*

*Proof.* We construct $V_{\pi^*}(\mathbf{s})$ by stepping through inductive steps in value iteration, and showing that the monotonicity property is preserved throughout these steps. We note that performing Value iteration from any initialization results in the optimal policy, $V_{\pi^*}(\mathbf{s})$

Let $Q^{(t)}(\mathbf{s}, \mathbf{a})$ represent the $Q$ value for iteration $t$ of Value iteration, and let $V^{(t)}(\mathbf{s})$ be the value function for iteration $t$. We initialize $V^{(0)}(\mathbf{s}) = 0$ for any $\mathbf{s}$, and note that the initialization should not impact the process. Running an iteration of value iteration leads $Q^{(1)}(\mathbf{s}, \mathbf{a}) = R(\mathbf{s}, \mathbf{a})$, so that $V^{(1)}(\mathbf{s}) = \max_{\mathbf{a}} Q_{\pi^*}^{(1)}(\mathbf{s}, \mathbf{a})$.

**Base Case**: Initially, $V^{(1)}(\mathbf{s}) = \max_{\mathbf{a}} R(\mathbf{s}, \mathbf{a})$. Next consider any two states, $\mathbf{x}$ and $\mathbf{y}$, so that $\mathbf{x} \geq \mathbf{y}$. Here $\mathbf{x} \geq \mathbf{y}$ implies that $x_i \geq y_i \forall i$. Let $\mathbf{a}^* = \text{argmax}_{\mathbf{a}} R(\mathbf{y}, \mathbf{a})$. Then $V^{(1)}(\mathbf{y}) = R(\mathbf{y}, \mathbf{a}^*) \leq R(\mathbf{x}, \mathbf{a}^*) \leq V^{(1)}(\mathbf{x})$. Therefore, $V^{(1)}(\mathbf{s})$ is monotonic.

**Inductive Hypothesis**: Assume that $V^{(t-1)}(\mathbf{s})$ is monotonic in $\mathbf{s}$.

Next, we will prove that $V^{(t)}(\mathbf{s} \vee \mathbf{e}_i) - V^{(t)}(\mathbf{s}) \geq 0$. Here $\mathbf{e}_i$ is the vector with 1 at location $i$ and 0 everywhere else.

Before proving this, we will first show that this implies monotonicity for any two states $\mathbf{s}, \mathbf{s}'$. Suppose that $V^{(t)}(\mathbf{s} \vee \mathbf{e}_i) - V^{(t)}(\mathbf{s}) \geq 0$. Then consider some pair of states, $\mathbf{s} \geq \mathbf{s}'$. Because $\mathcal{S} = \mathcal{A} = \{0, 1\}$, if $\mathbf{s} \geq \mathbf{s}'$, then we can write $\mathbf{s} = \mathbf{s}' \vee \bigvee_{i \in X} \mathbf{e}_i$ for some set $X = \{x_1, x_2, \cdots, x_n\}$. If $V^{(t)}(\mathbf{s}' \vee \mathbf{e}_i) \geq V^{(t)}(\mathbf{s}')$, then $V^{(t)}((\mathbf{s}' \vee \mathbf{e}_{x_1}) \vee \mathbf{e}_i) \geq V^{(t)}(\mathbf{s}' \vee \mathbf{e}_{x_1})$

Therefore, this implies that

$$V^{(t)}(\mathbf{s}) - V^{(t)}(\mathbf{s}') = \sum_{i=1}^{n} (V^{(t)}(\mathbf{s}' \vee \bigvee_{j=1}^{i} \mathbf{e}_{x_j}) - V^{(t)}(\mathbf{s}' \vee \bigvee_{j=1}^{i-1} \mathbf{e}_{x_j})) \geq 0 \qquad (10)$$

We get this by expanding $V^{(t)}(\mathbf{s}) - V^{(t)}(\mathbf{s}')$ into a telescoping series. Because each term in the sum is non-negative, the whole sum is non-negative.

Next we prove our original statement: $V^{(t)}(\mathbf{s} + \mathbf{e}_i) \geq V^{(t)}(\mathbf{s})$. We first expand $V^{(t)}(\mathbf{s}) = R(\mathbf{s}, \mathbf{a}) + \gamma \sum_{\mathbf{s}'} P(\mathbf{s}, \mathbf{a}, \mathbf{s}') V^{(t-1)}(\mathbf{s}')$ for some $\mathbf{a}$, where $P(\mathbf{s}, \mathbf{a}, \mathbf{s}') = \prod_{j=1}^{N} P_j(s_j, a_j, s_j')$. We note that $V^{(t)}(\mathbf{s} \vee \mathbf{e}_i) \geq R(\mathbf{s} \vee \mathbf{e}_i, \mathbf{a}) + \gamma \sum_{\mathbf{s}'} P(\mathbf{s} \vee \mathbf{e}_i, \mathbf{a}, \mathbf{s}') V^{(t-1)}(\mathbf{s}')$

Therefore:

$$V^{(t)}(\mathbf{s} \vee \mathbf{e}_i) - V^{(t)}(\mathbf{s}) \geq R(\mathbf{s} \vee \mathbf{e}_i, \mathbf{a}) - R(\mathbf{s}, \mathbf{a}) + \gamma \sum_{\mathbf{s}'} (P(\mathbf{s} \vee \mathbf{e}_i, \mathbf{a}, \mathbf{s}') - P(\mathbf{s}, \mathbf{a}, \mathbf{s}')) V^{(t-1)}(\mathbf{s}') \quad (11)$$

We note that $R(\mathbf{s} \vee \mathbf{e}_i, \mathbf{a}) - R(\mathbf{s}, \mathbf{a}) \geq 0$ due to the monotonicity of $R$ with fixed $\mathbf{a}$.

Let $\mathcal{C}_i \subset \{0,1\}^N$ be the set of 0-1 vectors of length $N$ so that if $x \in \mathcal{C}_i$ then $x_i = 0$. Additionally, let $\delta_i = P(\mathbf{s} \vee \mathbf{e}_i, \mathbf{a}, \mathbf{s}' \vee \mathbf{e}_i) - P(\mathbf{s}, \mathbf{a}, \mathbf{s}' \vee \mathbf{e}_i)$. We note that

$$
\begin{aligned}
\delta_i & \\
&= P(\mathbf{s} \vee \mathbf{e}_i, \mathbf{a}, \mathbf{s}' \vee \mathbf{e}_i) - P(\mathbf{s}, \mathbf{a}, \mathbf{s}' \vee \mathbf{e}_i) \\
&= \prod_{k=1}^N P_k((\mathbf{s} \vee \mathbf{e}_i)_k, a_k, (\mathbf{s}' \vee \mathbf{e}_i)_k) - \prod_{k=1}^N P_k(s_k, a_k, (\mathbf{s}' \vee \mathbf{e}_i)_k) \\
&= (P_i(1, a_i, 1) - P_i(s_i, a_i, 1)) \prod_{k=1, k \neq i}^N P_k(s_k, a_k, s_k') \\
&\geq 0
\end{aligned}
$$

Let $\mathbf{1}_i$ be the vector of all 1s except at $i$. Let $\wedge$ be the element-wise minimum operator. Then $\mathbf{s} \wedge \mathbf{1}_i$ sets $s_i = 0$ while leaving other indices unchanged. Because transition probabilities add to one, $P(\mathbf{s} \vee \mathbf{e}_i, \mathbf{a}, \mathbf{s}' \wedge \mathbf{1}_i) - P(\mathbf{s}, \mathbf{a}, \mathbf{s}' \wedge \mathbf{1}_i) = -\delta_i$

Next:

$V^{(t)}(\mathbf{s} \vee \mathbf{e}_i) - V^{(t)}(\mathbf{s})$

$$
\begin{aligned}
&\geq R(\mathbf{s} \vee \mathbf{e}_i, \mathbf{a}) - R(\mathbf{s}, \mathbf{a}) + \gamma \sum_{\mathbf{s}'} (P(\mathbf{s} \vee \mathbf{e}_i, \mathbf{a}, \mathbf{s}') - P(\mathbf{s}, \mathbf{a}, \mathbf{s}')) V^{(t-1)}(\mathbf{s}') \\
&\geq \gamma \sum_{\mathbf{s}'} (P(\mathbf{s} \vee \mathbf{e}_i, \mathbf{a}, \mathbf{s}') - P(\mathbf{s}, \mathbf{a}, \mathbf{s}')) V^{(t-1)}(\mathbf{s}') \\
&= \gamma \sum_{\mathbf{s}' \in \mathcal{C}_i} (P(\mathbf{s} \vee \mathbf{e}_i, \mathbf{a}, \mathbf{s}' \wedge \mathbf{1}_i) - P(\mathbf{s}, \mathbf{a}, \mathbf{s}' \wedge \mathbf{1}_i)) V^{(t-1)}(\mathbf{s}' \wedge \mathbf{1}_i) \\
&\quad + \gamma \sum_{\mathbf{s}' \in \mathcal{C}_i} (P(\mathbf{s} \vee \mathbf{e}_i, \mathbf{a}, \mathbf{s}' \vee \mathbf{e}_i) - P(\mathbf{s}, \mathbf{a}, \mathbf{s}' \vee \mathbf{e}_i)) V^{(t-1)}(\mathbf{s}' \vee \mathbf{e}_i) \\
&= \gamma \sum_{\mathbf{s}' \in \mathcal{C}_i} -\delta_i V^{(t-1)}(\mathbf{s}' \wedge \mathbf{1}_i) + \delta_i V^{(t-1)}(\mathbf{s}' \vee \mathbf{e}_i) \\
&= \gamma \sum_{\mathbf{s}' \in \mathcal{C}_i} \delta_i (V^{(t-1)}(\mathbf{s}' \vee \mathbf{e}_i) - V^{(t-1)}(\mathbf{s}' \wedge \mathbf{1}_i)) \\
&\geq 0
\end{aligned}
$$

The second step splits the sum into transitions for states with $s_i' = 0$ and those with $s_i' = 1$. The final step applies the inductive hypothesis and noting that $\mathbf{s}' \vee \mathbf{e}_i \geq \mathbf{s}' \wedge \mathbf{1}_i$. $\qquad \square$

**Lemma 10.** *Let $\mathcal{P}$ be a set of transition functions, $P_1, \cdots, P_N \in \mathcal{S} \times \mathcal{A} \times \mathcal{S} \to [0,1]$, where $N$ is the number of arms. Let $\mathcal{S} = \mathcal{A} = \{0,1\}$. Let $Q_\pi(\mathbf{s}, \mathbf{a}) = \mathbb{E}_{(\mathbf{s}', \mathbf{a}') \sim (\mathcal{P}, \pi)}[\sum_{t=0}^\infty \gamma^t R(\mathbf{s}', \mathbf{a}'))| \mathbf{s_0} = \mathbf{s}, \mathbf{a_0} = \mathbf{a}]$ for some policy $\pi : \mathcal{S} \to \mathcal{A}$, and let $\pi^*$ be the optimal policy. Let $V_{\pi^*}(\mathbf{s}) = \max_{\mathbf{a}} Q_\pi(\mathbf{s}, \mathbf{a})$. If $R(\mathbf{s}, \mathbf{a})$ is submodular in both $\mathbf{s}$ and $\mathbf{a}$, $g(\mathbf{s}) = \max_{\mathbf{a}} R(\mathbf{s}, \mathbf{a})$ is submodular in $\mathbf{s}$, and $P_i(1, a, 1) \geq P_i(0, a, 1)$ for any $a$, then $V_{\pi^*}(\mathbf{s})$ is submodular in $\mathbf{s}$.*

*Proof.* We similarly construct $V_{\pi^*}(\mathbf{s})$ through induction in the value iteration steps.

**Base Case:** We start with the base case: $V^{(1)}(\mathbf{s}) = \max_{\mathbf{a}} R(\mathbf{s}, \mathbf{a}) = g(\mathbf{s})$, which is submodular by assumption.

**Inductive Hypothesis:** Again let $V^{(t)}(\mathbf{s})$ be the value function at step $t$ of value iteration. We assume that $V^{(t-1)}(\mathbf{s})$ is submodular, and aim to prove that $V^{(t)}(\mathbf{s} \vee \mathbf{e}_i) - V^{(t)}(\mathbf{s}) - V^{(t)}(\mathbf{s} \vee \mathbf{e}_i \vee \mathbf{e}_j) + V^{(t)}(\mathbf{s} \vee \mathbf{e}_j) \geq 0$. We note that this is an equivalent definition of submodularity [37].

Now, consider $V^{(t)}(\mathbf{s} \vee \mathbf{e}_i) - V^{(t)}(\mathbf{s}) - V^{(t)}(\mathbf{s} \vee \mathbf{e}_i \vee \mathbf{e}_j) + V^{(t)}(\mathbf{s} \vee \mathbf{e}_j)$. The sum of submodular functions is submodular and note that $V^{(t)}(\mathbf{s}) = \max_{\mathbf{a}} R(\mathbf{s}, \mathbf{a}) + \gamma \sum_{\mathbf{s}'} V^{(t-1)}(\mathbf{s}') P(\mathbf{s}, \mathbf{a}, \mathbf{s}')$. We

note that the first term is submodular by assumption, so we focus on proving the submodularity of $\gamma \sum_{\mathbf{s}'} V^{(t-1)}(\mathbf{s}')P(\mathbf{s}, \mathbf{a}, \mathbf{s}')$. Let $\mathcal{C}_{i,j} \subset \{0,1\}^N$ be the set of 0-1 vectors of length $N$ so that if $x \in \mathcal{C}_{i,j}$ then $x_i = 0, x_j = 0$, and let $\delta_i = P_i(1, a_i, 1) - P_i(s_i, a_i, 1)$; note the different definition of $\delta_i$ from the previous proof. Finally, let $C(\mathbf{s}, \mathbf{a}, \mathbf{s}') = \prod_{l=1, l\neq i,j}^{N} P_l(s_l, a_l, s_l')$, then we can write the following:

$$V^{(t)}(\mathbf{s} \vee \mathbf{e}_i) - V^{(t)}(\mathbf{s}) - V^{(t)}(\mathbf{s} \vee \mathbf{e}_i \vee \mathbf{e}_j) + V^{(t)}(\mathbf{s} \vee \mathbf{e}_j)$$

$$= \gamma \sum_{\mathbf{s}' \in \mathcal{C}_{i,j}} C(\mathbf{s}, \mathbf{a}, \mathbf{s}')(V^{(t-1)}(\mathbf{s}' \vee \mathbf{e}_i) - V^{(t-1)}(\mathbf{s}'))\delta_i P_j(s_j, a_j, 0)$$

$$+ \gamma \sum_{\mathbf{s}' \in \mathcal{C}_{i,j}} C(\mathbf{s}, \mathbf{a}, \mathbf{s}')(V^{(t-1)}(\mathbf{s}' \vee \mathbf{e}_i \vee \mathbf{e}_j) - V^{(t-1)}(\mathbf{s}' \vee \mathbf{e}_j))\delta_i(1 - P_j(s_j, a_j, 0))$$

$$- \gamma \sum_{\mathbf{s}' \in \mathcal{C}_{i,j}} C(\mathbf{s}, \mathbf{a}, \mathbf{s}')(V^{(t-1)}(\mathbf{s}' \vee \mathbf{e}_i) - V^{(t-1)}(\mathbf{s}'))\delta_i(P_j(1, a_j, 0))$$

$$- \gamma \sum_{\mathbf{s}' \in \mathcal{C}_{i,j}} C(\mathbf{s}, \mathbf{a}, \mathbf{s}')(V^{(t-1)}(\mathbf{s}' \vee \mathbf{e}_i \vee \mathbf{e}_j) - V^{(t-1)}(\mathbf{s}' \vee \mathbf{e}_j))\delta_i(1 - P_j(1, a_j, 0))$$

$$= \gamma \sum_{\mathbf{s}' \in \mathcal{C}_{i,j}} C(\mathbf{s}, \mathbf{a}, \mathbf{s}')\delta_i(V^{(t-1)}(\mathbf{s}' \vee \mathbf{e}_i) - V^{(t-1)}(\mathbf{s}'))(P_j(s_j, a_j, 0) - P_j(1, a_j, 0))$$

$$+ \gamma \sum_{\mathbf{s}' \in \mathcal{C}_{i,j}} C(\mathbf{s}, \mathbf{a}, \mathbf{s}')\delta_i(V^{(t-1)}(\mathbf{s}' \vee \mathbf{e}_i \vee \mathbf{e}_j) - V^{(t-1)}(\mathbf{s}' \vee \mathbf{e}_j))(P_j(1, a_j, 0) - P_j(s_j, a_j, 0))$$

$$= \gamma \sum_{\mathbf{s}' \in \mathcal{C}_{i,j}} C(\mathbf{s}, \mathbf{a}, \mathbf{s}')\delta_i\delta_j(V^{(t-1)}(\mathbf{s}' \vee \mathbf{e}_i) + V^{(t-1)}(\mathbf{s}' \vee \mathbf{e}_i) - V^{(t-1)}(\mathbf{s}') - V^{(t-1)}(\mathbf{s}' \vee \mathbf{e}_i \vee \mathbf{e}_j))$$

$$\geq 0$$

The last step arises because $\delta_i$, $C(\mathbf{s}, \mathbf{a}, \mathbf{s}')$ are all non-negative, and the inductive hypothesis. $\square$

**Theorem 3.** *Consider an* RMAB-G *instance* $(\mathcal{S}, \mathcal{A}, R_i, R_{\text{glob}}, P_i, \gamma)$. *Let* $Q_\pi(\mathbf{s}, \mathbf{a}) = \mathbb{E}_{(\mathbf{s}', \mathbf{a}') \sim (\mathcal{P}, \pi)}[\sum_{t=0}^{\infty} \gamma^t R(\mathbf{s}', \mathbf{a}'))|\mathbf{s}^{(0)} = \mathbf{s}, \mathbf{a}^{(0)} = \mathbf{a}]$ *and let* $\pi^*$ *be the optimal policy. Let* $\mathbf{a}^* = \arg\max_{\mathbf{a}} Q_{\pi^*}(\mathbf{s}, \mathbf{a})$. *Suppose* $P_i(s, 1, 1) \geq P_i(s, 0, 1)\forall s, i$, $P_i(1, a, 1) \geq P_i(0, a, 1)\forall i, a$, $R(\mathbf{s}, \mathbf{a})$ *monotonic and submodular in* $\mathbf{s}$ *and* $\mathbf{a}$, *and* $g(\mathbf{s}) = \max_{\mathbf{a}} R(\mathbf{s}, \mathbf{a})$ *submodular in* $\mathbf{s}$. *Then with oracle access to* $Q_{\pi^*}$ *we can compute* $\hat{\mathbf{a}}$ *in* $\mathcal{O}(N^2)$ *time so* $Q_{\pi^*}(\mathbf{s}, \hat{\mathbf{a}}) \geq (1 - \frac{1}{e})Q_{\pi^*}(\mathbf{s}, \mathbf{a}^*)$.

*Proof.* We first demonstrate that $Q_{\pi^*}(\mathbf{s}, \mathbf{a})$ is submodular and monotonic in $\mathbf{a}$. Prior work demonstrates that monotonic submodular function can be approximated within an approximation factor of $1 - \frac{1}{e}$ with $\mathcal{O}(N^2)$ evaluations of the function, so all that remains is to demonstrate submodularity [13].

We note that $Q_{\pi^*}(\mathbf{s}, \mathbf{a}) = R(\mathbf{s}, \mathbf{a}) + \gamma \sum_{\mathbf{s}'} V_{\pi^*}(\mathbf{s}')P(\mathbf{s}, \mathbf{a}, \mathbf{s}')$, and that $R(\mathbf{s}, \mathbf{a})$ is both monotonic and submodular in $\mathbf{a}$. Therefore, we demonstrate that $\gamma \sum_{\mathbf{s}'} V_{\pi^*}(\mathbf{s}')P(\mathbf{s}, \mathbf{a}, \mathbf{s}')$ is monotonic and submodular in $\mathbf{a}$. Let $H(\mathbf{a}) = \gamma \sum_{\mathbf{s}'} V_{\pi^*}(\mathbf{s}')P(\mathbf{s}, \mathbf{a}, \mathbf{s}')$ for fixed $\mathbf{s}$.

We start by demonstrating monotonicity. First, let $\eta_i = P_i(s_i, 1, 1) - P_i(s_i, 0, 1)$, and by assumption, $\eta_i \geq 0$. Then:

$$H(\mathbf{a} \vee \mathbf{e}_i) - H(\mathbf{a})$$

$$= \gamma \sum_{\mathbf{s}' \in \mathcal{C}_i} V_{\pi^*}(\mathbf{s} \vee \mathbf{e}_i)(P(\mathbf{s}, \mathbf{a} \vee \mathbf{e}_i, \mathbf{s} \vee \mathbf{e}_i) - P(\mathbf{s}, \mathbf{a}, \mathbf{s} \vee \mathbf{e}_i))$$

$$+ \gamma \sum_{\mathbf{s}' \in \mathcal{C}_i} V_{\pi^*}(\mathbf{s} \vee \mathbf{e}_i)(P(\mathbf{s}, \mathbf{a} \vee \mathbf{e}_i, \mathbf{s} \wedge \mathbf{1}_i) - P(\mathbf{s}, \mathbf{a}, \mathbf{s} \wedge \mathbf{1}_i))$$

$$= \gamma \sum_{\mathbf{s}' \in \mathcal{C}_i} \eta_i(V_{\pi^*}(\mathbf{s} \vee \mathbf{e}_i) - V_{\pi^*}(\mathbf{s} \wedge \mathbf{1}_i)) \prod_{j=1, j\neq i}^{N} P_j(s_j, a_j, s_j')$$

$$\geq 0$$

The last steps follows from the monotonicity of $V_{\pi^*}(\mathbf{s})$, which was proven in Lemma 9.

Next, we prove submodularity of $H(\mathbf{a})$, by showing that $H(\mathbf{a} \vee \mathbf{e}_i) - H(\mathbf{a}) - H(\mathbf{a} \vee \mathbf{e}_i \vee \mathbf{e}_j) + H(\mathbf{a} \vee \mathbf{e}_j) \geq 0$. That is

$$
\begin{aligned}
&H(\mathbf{a} \vee \mathbf{e}_i) - H(\mathbf{a}) - H(\mathbf{a} \vee \mathbf{e}_i \vee \mathbf{e}_j) + H(\mathbf{a} + \vee \mathbf{e}_j) \\
&= \gamma \sum_{\mathbf{s}' \in \mathcal{C}_{i,j}} \eta_i P_j(s_j, a_j, 0)(V_{\pi^*}(\mathbf{s} \vee \mathbf{e}_i) - V_{\pi^*}(\mathbf{s})) + \eta_i(1 - P_j(s_j, a_j, 0))(V_{\pi^*}(\mathbf{s} \vee \mathbf{e}_i \vee \mathbf{e}_j) - V_{\pi^*}(\mathbf{s})) \\
&\quad - \gamma \sum_{\mathbf{s}' \in \mathcal{C}_{i,j}} \eta_i P_j(s_j, 1, 0)(V_{\pi^*}(\mathbf{s} \vee \mathbf{e}_i) - V_{\pi^*}(\mathbf{s})) + \eta_i(1 - P_j(s_j, 1, 0))(V_{\pi^*}(\mathbf{s} \vee \mathbf{e}_i \vee \mathbf{e}_j) - V_{\pi^*}(\mathbf{s} \vee \mathbf{e}_i)) \\
&= \gamma \sum_{\mathbf{s}' \in \mathcal{C}_{i,j}} \eta_i \eta_j (V_{\pi^*}(\mathbf{s} \vee \mathbf{e}_i) + V_{\pi^*}(\mathbf{s} \vee \mathbf{e}_j) - V_{\pi^*}(\mathbf{s}) - V(\mathbf{s} \vee \mathbf{e}_i \vee \mathbf{e}_j)) \\
&\geq 0
\end{aligned}
$$

The last line follows from the submodularity of $V$, as proven in Lemma 10. Therefore, we have shown that $Q_{\pi^*}(\mathbf{s}, \mathbf{a})$ is submodular, which completes the proof. $\qquad\square$

# L  Proofs of Index-Based Bounds

## L.1  Lower Bounds

**Theorem 4.** *For any fixed set of transitions, $\mathcal{P}$, let $\mathrm{OPT} = \max_\pi \mathbb{E}_{(\mathbf{s},\mathbf{a}) \sim (P,\pi)}[\frac{1}{T} \sum_{t=0}^{T-1} R(\mathbf{s}, \mathbf{a})]$ for some $T$. For an RMAB-G $(\mathcal{S}, \mathcal{A}, R_i, R_{\mathrm{glob}}, P_i, \gamma)$, let $R_i'(s_i, a_i) = R_i(s_i, a_i) + p_i(s_i)a_i$, and let the induced linear RMAB be $(\mathcal{S}, \mathcal{A}, R_i', P_i, \gamma)$. Let $\pi_{\mathrm{linear}}$ be the Linear-Whittle policy, and let $\mathrm{ALG} = \mathbb{E}_{(\mathbf{s},\mathbf{a}) \sim (P,\pi_{\mathrm{linear}})}[\frac{1}{T} \sum_{t=0}^{T-1} R(\mathbf{s}, \mathbf{a})]$. Define $\beta_{\mathrm{linear}}$ as*

$$
\beta_{\mathrm{linear}} = \min_{\mathbf{s} \in \mathcal{S}^N, \mathbf{a} \in [0,1]^N, \|\mathbf{a}\|_1 \leq K} \frac{R(\mathbf{s}, \mathbf{a})}{\sum_{i=1}^N (R_i(s_i, a_i) + p_i(s_i)a_i)} \tag{12}
$$

*If the induced linear RMAB is irreducible and indexable with the uniform global attractor property, then $\mathrm{ALG} \geq \beta_{\mathrm{linear}}\mathrm{OPT}$ asymptotically in $N$ for any set of transitions, $\mathcal{P}$.*

*Proof.* By the definition of the minimum, $\beta_{\mathrm{linear}} \sum_{i=1}^N (p_i(s_i)a_i + R_i(s_i, a_i)) \leq R(\mathbf{s}, \mathbf{a}) \forall \mathbf{a}, \mathbf{s}$.

Additionally, note that $\sum_{i=1}^N p_i(s_i)a_i \geq R_{\mathrm{glob}}(\mathbf{s}, \mathbf{a})$. This holds because $p_i(s_i) = \max_{\mathbf{s}'|s_i'=s_i} R_{\mathrm{glob}}(\mathbf{s}, \mathbf{a})$. So, $\sum_{i=1}^N p_i(s_i)a_i \geq R(\mathbf{s}, \sum_{i=1}^N \mathbf{e}_i a_i) = R(\mathbf{s}, \mathbf{a})$ by submodularity of $R_{\mathrm{glob}}(\mathbf{s}, \mathbf{a})$.

We compare the linearized reward to the submodular reward for the set of actions played by the Linear-Whittle policy $\pi_{\mathrm{linear}}$. Let the actions be $\mathbf{a}^{(1)}, \mathbf{a}^{(2)}, \cdots$. If we play for $T$ iterations, then:

$$
\frac{\beta_{\mathrm{linear}}}{T} \sum_{t=0}^{T-1} \sum_{i=1}^N (p_i(s_i^{(t)})a_i^{(t)} + R_i(s_i^{(t)}, a_i^{(t)})) \leq \frac{1}{T} \sum_{t=0}^{T-1} R(\mathbf{s}^{(t)}, \mathbf{a}^{(t)}) \tag{13}
$$

Next, consider the following induced RMAB with $R_i'(s_i, a_i) = R_i(s_i, a_i) + p_i(s_i)a_i$. By assumption, our relaxed RMAB is indexable, irreducible, and has a global attractor. In this scenario, prior work [2] states that the Whittle index is asymptotically optimal.

For this relaxed RMAB problem, we note that the Whittle Index solution corresponds exactly to the Linear-Whittle policy, $\pi_{\mathrm{linear}}$. That is

$$
\max_\pi \mathbb{E}_{(\mathbf{s},\mathbf{a}) \sim (P,\pi)}\Big[\frac{1}{T} \sum_{t=0}^{T-1} \sum_{i=1}^N R_i'(s_i, a_i)\Big] = \mathbb{E}_{(\mathbf{s},\mathbf{a}) \sim (P,\pi_{\mathrm{linear}})}\Big[\frac{1}{T} \sum_{t=0}^{T-1} \sum_{i=1}^N R_i'(s_i, a_i)\Big] \tag{14}
$$

Then, asymptotically,

$$\text{OPT}$$

$$= \max_\pi \mathbb{E}_{(\mathbf{s},\mathbf{a})\sim(P,\pi)}[\frac{1}{T}\sum_{t=0}^{T-1} R(\mathbf{s},\mathbf{a})]$$

$$\leq \max_\pi \mathbb{E}_{(\mathbf{s},\mathbf{a})\sim(P,\pi)}[\frac{1}{T}\sum_{t=0}^{T-1}\sum_{i=1}^{N} R_i'(s_i,a_i)] = \mathbb{E}_{(\mathbf{s},\mathbf{a})\sim(P,\pi_{\text{linear}})}[\frac{1}{T}\sum_{t=0}^{T-1}\sum_{i=1}^{N} R_i'(s_i,a_i)]$$

$$\leq \frac{1}{\beta_{\text{linear}}}\mathbb{E}_{(\mathbf{s},\mathbf{a})\sim(P,\pi_{\text{linear}})}[\frac{1}{T}\sum_{t=0}^{T-1} R(\mathbf{s},\mathbf{a})] = \frac{1}{\beta_{\text{linear}}}\text{ALG}$$

$\square$

**Theorem 5.** *For any fixed set of transitions, $\mathcal{P}$, let $\text{OPT} = \max_\pi \mathbb{E}_{(\mathbf{s},\mathbf{a})\sim(P,\pi)}[\frac{1}{T}\sum_{t=0}^{T-1} R(\mathbf{s},\mathbf{a})]$ for some $T$. For an* RMAB-G *$(\mathcal{S},\mathcal{A},R_i,R_{\text{glob}},P_i,\gamma)$, let $R_i'(s_i,a_i) = R_i(s_i,a_i) + u_i(s_i)a_i$, and let the induced Shapley* RMAB *be $(\mathcal{S},\mathcal{A},R_i',P_i,\gamma)$. Let $\pi_{\text{shapley}}$ be the Shapley-Whittle policy, and let $\text{ALG} = \mathbb{E}_{(\mathbf{s},\mathbf{a})\sim(P,\pi_{\text{shapley}})}[\frac{1}{T}\sum_{t=0}^{T-1} R(\mathbf{s},\mathbf{a})]$. Define $\beta_{\text{shapley}}$ as*

$$\beta_{\text{shapley}} = \frac{\min\limits_{\mathbf{s}\in\mathcal{S}^N,\mathbf{a}\in[0,1]^N,\|\mathbf{a}\|_1\leq K}\frac{R(\mathbf{s},\mathbf{a})}{\sum_{i=1}^N (R_i(s_i,a_i)+u_i(s_i)a_i)}}{\max\limits_{\mathbf{s}\in\mathcal{S}^N,\mathbf{a}\in[0,1]^N,\|\mathbf{a}\|_1\leq K}\frac{R(\mathbf{s},\mathbf{a})}{\sum_{i=1}^N (R_i(s_i,a_i)+u_i(s_i)a_i)}} \tag{15}$$

*If the induced Shapley* RMAB *is irreducible and indexable with the uniform global attractor property, then $\text{ALG} \geq \beta_{\text{shapley}}\text{OPT}$ asymptotically in $N$ for any choice of transitions, $\mathcal{P}$.*

*Proof.* First, consider $\text{OPT} = \max_\pi \mathbb{E}_{(\mathbf{s},\mathbf{a})\sim(P,\pi)}[\sum_{t=0}^{\infty}\gamma^t R(\mathbf{s},\mathbf{a})]$. First, note that $R(\mathbf{s}',\mathbf{a}') \leq (\sum_{i=1}^N u_i(s_i')a_i' + R_i(s_i',a_i'))\max\limits_{\mathbf{s}\in\mathcal{S}^N,\mathbf{a}\in[0,1]^N,\|\mathbf{a}\|_1\leq K}\frac{R(\mathbf{s},\mathbf{a})}{\sum_{i=1}^N (u_i(s_i)a_i+R_i(s_i,a_i))}$. Next, note that similar to Theorem 4, $\pi_{\text{shapley}}$ is the optimal policy for the RMAB instance where $R_i'(s_i,a_i) = u_i(s_i)a_i + R_i(s_i,a_i)$ for the induced RMAB (due to the indexability, irreducibility, and global attractor properties [2]), or in other words,

$$\max_\pi \mathbb{E}_{(\mathbf{s},\mathbf{a})\sim(P,\pi)}[\frac{1}{T}\sum_{t=0}^{T-1}\sum_{i=1}^{N} R_i'(s_i,a_i)] = \mathbb{E}_{(\mathbf{s},\mathbf{a})\sim(P,\pi_{\text{shapley}})}[\frac{1}{T}\sum_{t=0}^{T-1}\sum_{i=1}^{N} R_i'(s_i,a_i)] \tag{16}$$

Therefore,

$$\text{OPT}$$

$$= \max_\pi \mathbb{E}_{(\mathbf{s},\mathbf{a})\sim(P,\pi)}[\frac{1}{T}\sum_{t=0}^{T-1} R(\mathbf{s},\mathbf{a})]$$

$$\leq \mathbb{E}_{(\mathbf{s},\mathbf{a})\sim(P,\pi)}[\frac{1}{T}\sum_{t=0}^{T-1}\sum_{i=1}^{N} R_i'(s_i,a_i)]\max\limits_{\mathbf{s}\in\mathcal{S}^N,\mathbf{a}\in[0,1]^N,\|\mathbf{a}\|_1\leq K}\frac{R(\mathbf{s},\mathbf{a})}{\sum_{i=1}^N (u_i(s_i)a_i+R_i(s_i,a_i))}$$

$$\leq \mathbb{E}_{(\mathbf{s},\mathbf{a})\sim(P,\pi_{\text{shapley}})}[\frac{1}{T}\sum_{t=0}^{T-1}\sum_{i=1}^{N} R_i'(s_i,a_i)]\max\limits_{\mathbf{s}\in\mathcal{S}^N,\mathbf{a}\in[0,1]^N,\|\mathbf{a}\|_1\leq K}\frac{R(\mathbf{s},\mathbf{a})}{\sum_{i=1}^N (u_i(s_i)a_i+R_i(s_i,a_i))} \leq \mathbb{E}_{(\mathbf{s},\mathbf{a})\sim(P,\pi_{\text{shapley}})}[\frac{1}{T}$$

$\square$

**Corollary 5.1.** *Consider an* RMAB-G *instance with a monotonic, submodular reward function $R(\mathbf{s},\mathbf{a}) = R_{\text{glob}}(\mathbf{s},\mathbf{a})$, and let $\pi_{\text{linear}}$ be the Linear-Whittle policy. Let $\text{ALG} = \mathbb{E}_{(\mathbf{s},\mathbf{a})\sim(P,\pi_{\text{linear}})}[\frac{1}{T}\sum_{t=0}^{T-1} R(\mathbf{s},\mathbf{a})]$ and $\text{OPT} = \max_\pi \mathbb{E}_{(\mathbf{s},\mathbf{a})\sim(P,\pi)}[\frac{1}{T}\sum_{t=0}^{T-1} R(\mathbf{s},\mathbf{a})]$ for some $T$. Then $\text{ALG} \geq \frac{\text{OPT}}{K}$ for any transitions $\mathcal{P}$.*

*Proof.* By Theorem 4, we know that $\text{ALG} \geq \beta_{\text{linear}}\text{OPT}$. We next show that $\beta_{\text{linear}} \geq \frac{1}{K}$.

For a monotonic, submodular function, $F : 2^\Omega \rightarrow \mathbb{R}$, $F(\{x_1, x_2, \cdots, x_n\}) \geq \max_i F(\{x_i\})$. Next, applying this to our global reward function, for a fixed state $\mathbf{s}$, we note that $R_{\text{glob}}(\mathbf{s}, \mathbf{a}) \geq \max_i R_{\text{glob}}(\mathbf{s}, \mathbf{e}_i a_i)$. Next, by construction, $R(\mathbf{s}, \mathbf{a}) = \sum_{i=1}^{N} R_i(s_i, a_i) + R_{\text{glob}}(\mathbf{s}, \mathbf{a}) \leq \sum_{i=1}^{N}(R_i(s_i, a_i) + p_i(s_i)a_i)$. Additionally, $R(\mathbf{s}, \mathbf{a}) \geq \sum_{i=1}^{N} R_i(s_i, a_i) + \max_i p_i(s_i)a_i$

Next, for any state $\mathbf{s}$, $\sum_{i=1}^{N} p_i(s_i)a_i \leq K \max_i p_i(s_i)a_i$.

Combining the bounds on $R(\mathbf{s}, \mathbf{a})$ and $\sum_{i=1}^{N} p_i(\mathbf{s}, \mathbf{a})$, gives that for any state $\mathbf{s}$,

$$\frac{R(\mathbf{s}, \mathbf{a})}{\sum_{i=1}^{N}(R_i(s_i, 0) + p_i(s_i)a_i)} \geq \frac{\sum_{i=1}^{N} R_i(s_i, a_i) + \max_i p_i(s_i)a_i}{\sum_{i=1}^{N} R_i(s_i, a_i) + K \max_i p_i(s_i)a_i} \geq \frac{1}{K} \tag{17}$$

Therefore, $\beta_{\text{linear}} \geq \frac{1}{K}$, completing our proof.

$\square$

## L.2 Upper Bounds

**Theorem 6.** *Let* $\hat{\mathbf{a}}(\mathbf{s}) = \underset{\mathbf{a}\in[0,1]^N, \|\mathbf{a}\|_1 \leq K}{\arg\max} \sum_{i=1}^{N}(R_i(s_i, a_i) + p_i(s_i)a_i)$. *For a set of transitions* $\mathcal{P} = \{P_1, P_2, \cdots, P_N\}$, *let* $\text{OPT} = \max_\pi \mathbb{E}_{(\mathbf{s}, \mathbf{a}) \sim (P, \pi)}[\sum_{t=0}^{\infty} \gamma^t R(\mathbf{s}, \mathbf{a})|\mathbf{s}^{(0)}]$ *and* $\text{ALG} = \mathbb{E}_{(\mathbf{s}, \mathbf{a}) \sim (P, \pi_{\text{linear}})}[\sum_{t=0}^{\infty} \gamma^t R(\mathbf{s}, \mathbf{a})|\mathbf{s}^{(0)}]$. *Define* $\theta_{\text{linear}}$ *as follows:*

$$\theta_{\text{linear}} = \min_{\mathbf{s}\in\mathcal{S}^N} \frac{R(\mathbf{s}, \hat{\mathbf{a}}(\mathbf{s}))}{\max_{\mathbf{a}\in[0,1]^N, \|\mathbf{a}\|_1 \leq K} R(\mathbf{s}, \mathbf{a})} \tag{18}$$

*Then there exists some transitions, $\mathcal{P}$, and initial state $\mathbf{s}^{(0)}$, so that* $\text{ALG} \leq \theta_{\text{linear}}\text{OPT}$

*Proof.* We prove this by constructing such a set of transitions such that $\text{ALG} \leq \theta_{\text{linear}}\text{OPT}$. Let $\mathbf{s}^{(0)} = \arg\min_{\mathbf{s}} \frac{R(\mathbf{s}, \hat{\mathbf{a}})}{\max_{\mathbf{a}\in[0,1]^N, \|\mathbf{a}\|_1 \leq K} R(\mathbf{s}, \mathbf{a})}$. Next, let $P_i$ be the identity transition; $P_i(s_i, a_i, s_i^{(0)}) = 1$ for any $\mathbf{s}, \mathbf{a}, i$. Under these transitions, all arms stay in the state $\mathbf{s}^{(0)}$ for all time periods $t$.

Because the state is constant, the set of actions played, $\mathbf{a}^{(t)}$ is constant across time periods (as the Linear-Whittle algorithm is deterministic). We next show that the Linear-Whittle policy plays the action $\underset{\mathbf{a}\in[0,1]^N, \|\mathbf{a}\|_1 \leq K}{\arg\max} \sum_{i=1}^{N}(p_i(s_i)a_i + R_i(s_i, a_i))$. We note that all arms have the same transitions; therefore, arms are played in order of $p_i(s_i^{(0)}) + R_i(s_i, a_i)$, as each arm is predicted to receive a reward $a_i p_i(s_i^{(0)}) + R_i(s_i, a_i)$. This is because Linear-Whittle aims to maximize the sum of marginal rewards, and when the transitions are homogeneous, chooses the largest values for $p_i(s_i)a_i + R_i(s_i, a_i)$. In other words, the Linear-Whittle policy chooses an action, $\mathbf{a}_{\text{linear}}$ so that

$$\mathbf{a}^{\text{linear}} = \underset{\mathbf{a}\in[0,1]^N, \|\mathbf{a}\|_1 \leq K}{\arg\max} \sum_{i=1}^{N}(p_i(s_i^{(0)})a_i + R_i(s_i, a_i)) = \hat{\mathbf{a}}(\mathbf{s}) \tag{19}$$

Finally, we observe that the optimal action to play is $\arg\max_{\mathbf{a}\in[0,1]^N, \|\mathbf{a}\|_1 \leq K} R(\mathbf{s}^{(0)}, \mathbf{a})$. The ratio of rewards between these two actions is exactly $\theta_{\text{linear}}$, and therefore, $\text{ALG} \leq \theta_{\text{linear}}\text{OPT}$ $\square$

**Theorem 7.** *Let* $\hat{\mathbf{a}}(\mathbf{s}) = \underset{\mathbf{a}\in[0,1]^N, \|\mathbf{a}\|_1 \leq K}{\arg\max} \sum_{i=1}^{N}(R_i(s_i, a_i) + u_i(s_i)a_i)$. *For a set of transitions* $\mathcal{P} = \{P_1, P_2, \cdots, P_N\}$ *and initial state* $\mathbf{s}^{(0)}$, *let* $\text{OPT} = \max_\pi \mathbb{E}_{(\mathbf{s}, \mathbf{a}) \sim (P, \pi)}[\sum_{t=0}^{\infty} \gamma^t R(\mathbf{s}, \mathbf{a})|\mathbf{s}^{(0)}]$ *and* $\text{ALG} = \mathbb{E}_{(\mathbf{s}, \mathbf{a}) \sim (P, \pi_{\text{shapley}})}[\sum_{t=0}^{\infty} \gamma^t R(\mathbf{s}, \mathbf{a})|\mathbf{s}^{(0)}]$. *Let* $\theta_{\text{shapley}}$ *be:*

$$\theta_{\text{shapley}} = \min_{\mathbf{s}} \frac{R(\mathbf{s}, \hat{\mathbf{a}}(\mathbf{s}))}{\max_{\mathbf{a}\in[0,1]^N, \|\mathbf{a}\|_1 \leq K} R(\mathbf{s}, \mathbf{a})} \tag{20}$$

*Then there exists some transitions, $\mathcal{P}$ and initial state $\mathbf{s}^{(0)}$ so that* $\mathrm{ALG} \leq \theta_{\mathrm{shapley}}\mathrm{OPT}$

*Proof.* We follow the same strategy as the proof for the Linear-Whittle case and construct a set of transition probabilities so that $\mathrm{ALG} \leq \theta_{\mathrm{shapley}}\mathrm{OPT}$. Again, let $\mathbf{s}^{(0)} = \mathrm{argmax}_{\mathbf{s}} \frac{R(\mathbf{s}, \hat{\mathbf{a}}(\mathbf{s}))}{\max\limits_{\mathbf{a} \in [0,1]^N, \|\mathbf{a}\|_1 \leq K} R(\mathbf{s}, \mathbf{a})}$
and $P_i$ be the identity transition. By the same reasoning as Theorem 6, the Shapley-Whittle policy will play the action $\hat{\mathbf{a}}$; the Linear- and Shapley-Whittle situations are analogous, with the only difference being that Shapley-Whittle assumes each arm receives a reward of $a_i u_i(s_i^{(0)})$ instead of $a_i p_i(s_i^{(0)})$. The optimal action played receives a reward of $\max\limits_{\mathbf{a} \in [0,1]^N, \|\mathbf{a}\|_1 \leq K} R(\mathbf{s}, \mathbf{a})$, so that the ratio between the reward of the Shapley-Whittle policy and optimal is

$$\min_{\mathbf{s} \in \mathcal{S}^N} \frac{R(\mathbf{s}, \hat{\mathbf{a}}(\mathbf{s}))}{\max\limits_{\mathbf{a} \in [0,1]^N, \|\mathbf{a}\|_1 \leq K} R(\mathbf{s}, \mathbf{a})} \tag{21}$$

This is exactly $\theta_{\mathrm{shapley}}$ when $\mathbf{s} = \mathbf{s}^{(0)}$. $\qquad\square$

**Lemma 8.** *Define an index-based policy as any policy that can be described through a function $g(s_i, P_i, R)$, such that in state $\mathbf{s}$, the policy, $\pi$ selects the $K$ largest values of $g(s_i, P_i, R)$, where such a function is evaluated for each arm. Then index-based algorithms will achieve a discounted reward approximation ratio no better than $\frac{1-\gamma}{1-\gamma^K}$*

*Proof.* Consider an $N = K$ arm system, with $\mathcal{S} = \{0, 1\}$. Suppose our reward is $R(\mathbf{s}, \mathbf{a}) = \max\limits_i s_i a_i$. Next, define the transition probabilities for all arms as follows: $P_i(s, 0, s) = 1$ and $P_i(s, 1, 0) = 1$; if an arm is pulled, then the arm will transition to state 0, and otherwise, the arm remains in its current state.

Then note that because of symmetry between the arms, $g(s_i, P_i, R)$ is identical, as $R$ is symmetric across arms, and $s_i$ and $P_i$ is the same across arms. Moreover, because there are $N = K$ arms, all arms will be selected in the first time step. Therefore, any index-based strategy will lead to a reward of 1. However, selecting arm $i$ in timestep $i$ instead leads to rewards in each of the $K$ timesteps, leading to a total reward of $\sum_{i=0}^{N-1} \gamma^i > 1$ when $N > 1$. Applying the geometric sum formula reveals that this is a $\frac{1-\gamma}{1-\gamma^K}$ approximation. $\qquad\square$

