# OpenReview forum: "Global Rewards in Restless Multi-Armed Bandits"
_NeurIPS.cc/2024/Conference — NeurIPS 2024 poster_

### Official Review · Reviewer_a47J · 2024-07-03

**Soundness:** 4
**Presentation:** 4
**Contribution:** 4
**Rating:** 7
**Confidence:** 4

**Summary:**

The authors study restless multi-arm bandits where we do not observe separate reward for individual arms and instead observe a global reward which is the sum of reward across arms. They propose Linear and Shapley-Whittle indices, which extend classical Whittle indices  (designed for settings where separate reward for individual arms are observed) into the global reward setting and establish approximation bounds. The proposed linear-Whittle policy computes marginal reward for each arm, and the proposed Shapley-Whittle policy computes Shapley values for each arm. To address nonlinear reward functions, the authors propose adaptive policies that takes into account inter-arm interactions. They combine MCTS with Whittle indices to look beyond greedy selection.

**Strengths:**

The authors prove the difficulty of finding solutions to RMABs with global reward through reduction from RAMB and reduction from submodular optimization. They establish performance bounds for index-based policies and include discussions on the takeaways of the bounds.

A comprehensive set of policies and baselines are compared in experimental evaluations. Results on both synthetic and real-world data showcase the strength of proposed approaches. The real-world setting nicely motivates the global reward RMAB problem.

**Weaknesses:**

For approximation bounds in section 4.2, the authors briefly describe proof techniques and implications of the results. It would strengthen the paper to elaborate more on technical novelties / contributions in the proofs.

The adaptive policies are stronger than pre-computed index-based policies, which have exhibit poor performance due to their inability to consider inter-arm interactions. However, the theoretical results focus on linear and Shapley-Whittle indices.

**Questions:**

Could the theoretical results be modified and applied to adaptive approaches, or have implications in adaptive approaches? If so, it would be nice to have a discussion in the paper.

The authors focus on scenarios where the global reward is monotonic and submodular in actions. Such rewards have diminishing returns as extra arms are pulled. Could the authors discuss potential ways to apply/modify proposed approaches to settings where the global reward is not monotonic and submodular in actions?  This is mentioned in limitations section, but it could improve the paper to at least point out potential ways to apply proposed methods in more general settings.

**Limitations:**

The authors discussed the limitations and impact of the work.

---

> ### Author Rebuttal · Authors · 2024-08-06
>
> Dear Reviewer a47J,
>
> We thank you for your comments and suggestions. These comments will greatly help improve our final manuscript. We appreciate that you find our empirical evaluation comprehensive, and find that our empirical results strengthen the validity of our policies. We are additionally glad that our real-world applications help motivate the RMAB-G problem. We provide answers to each of your questions below:
>
> ### Novelties + Contributions in Approximation Bounds
>
> We thank the reviewers for inquiring about our proof techniques and the novelties present there. Our key technical insight is a novel method to bound the performance of Linear- and Shapely-Whittle policies using similarities between the global rewards and the rewards of an induced RMAB. Such a technique provides insight into how approximation guarantees can be derived for other variants of RMABs. To quantify the similarity between the rewards of Linear- and Shapley-Whittle and the rewards of the induced RMAB, we use properties of submodular functions to upper bound the reward from Linear- and Shapley-Whittle. We plan to emphasize these contributions more in the camera-ready version of our paper.
>
> ### Non-Monotonic Rewards
>
> We thank the reviewers for bringing up this interesting question. Following the reviewer's suggestion, we empirically tested our methods with non-monotonic. In the supplemental PDF, we attach a Figure that evaluates the performance of our policies under two non-monotonic rewards. The first reward is the minimum function, where we assume that the minimum of an empty set is 0, and the second is the linear reward, but with negative rewards for some arms. We compare the performance of our policies for both 4 and 10 arms, and evaluate this across 15 seeds.
>
> MCTS Shapley-Whittle performs best for the negative linear reward, which follows the same trend seen for monotonic rewards. Such results follow intuition, as for non-monotonic rewards, the impact of one arm upon the rewards of other arms is more complex, requiring the use of adaptive algorithms. For the minimum function, we find that MCTS-based methods perform best for N=10, while for N=4, we find that our index-based policies perform near-optimally. We believe that a more systematic investigation of non-monotonic rewards poses an interesting direction for future research, and thank the reviewer for bringing this up.

---

> > ### Comment · Reviewer_a47J · 2024-08-13
> >
> > I thank the authors for the response. The authors nicely summarized the novelties and contribution in approximation bounds, which improves the presentation. The authors also pointed to empirical results in appendix across various settings under monotonic rewards, which improves the contribution. I increased the confidence in assessment of this work.

---

### Official Review · Reviewer_t2ns · 2024-07-05

**Soundness:** 3
**Presentation:** 2
**Contribution:** 3
**Rating:** 6
**Confidence:** 3

**Summary:**

This paper studies RMAB (restless multi-armed bandits) with global
rewards. Standard RMAB assumes that the total reward is a sum of the
rewards across arms. Even though this type of RMAB incurs
curse-of-dimensionality, the linear reward structure allows Whittle
index to be defined. In contrast, in this paper the total reward has an
additional term that is submodular but non-linear. The authors first
propose two linear approximations of the global reward, through which
the Whittle policy can then to used. The approximation bounds of the
resulting linear-Whittle policy and Shapley-Whittle policy are provided.
Since these approximation bounds can be poor, the authors further
provide an iterative linear approximation (iterative linear-Whittle
policy), although the latter does not have performance guarantees.

**Strengths:**

1. RMAB with global reward has not been studied in the literature.

2. Some approximation guarantees are provided for the Linear-Whittle and
Shapley-Linear Whittle policies.

**Weaknesses:**

1. The provable approximation guarantees can be loose (as low as $1/K$,
where $K$ is the number of arms), unless the non-linear term is
insignificant.

2. The iterative linear-Whittle policy, which is meant to achieve better
performance, does not have performance guarantees.

3. The presentation of the iterative linear-Whittle policy could be
improved.

**Questions:**

1. What is the significance of Theorem 3? The reviewer is confused
because only ensuring that $\hat{a}$ attains a certain fraction of the
optimal $Q$-function does not seem to imply that the long-term reward is
guaranteed to be lower bounded.

Also, it seems that in this theorem the argument of $P_i$ for the state
is either 0 or 1. Why are there only two states?

2. In Example 4.1, how do you know that the approximation ratio is 1/2
by only examining one $a$ and $s$. Shouldn't you take the minimum over
all $a$ and $s$?

3. The Definition 4 (Iterative Linear-Whittle) is a bit confusing. The
iterations were never introduced. It seems that $X$ is the decision
from the previous iteration, is that right? It would be better to state
the algorithm in a pseudo code with iterations.

4. Similarly, Section 5.3 is difficult to follow. It is unclear to the
reader where/how/why Monte Carlo Tree Search is combined with the
iterative algorithm in Definition 4.

**Limitations:**

Limitations are discussed in Section 8.

---

> ### Author Rebuttal · Authors · 2024-08-06
>
> Dear Reviewer t2ns,
>
> We thank you for your suggestions and comments on our paper, and these will greatly help improve our final manuscript. We are glad that you appreciate how RMAB-G has not been previously studied in the literature. Additionally, we are happy that you appreciate the approximation guarantees we provide for our Linear- and Shapley-Whittle policies. We provide answers to each of your questions below:
>
> ### Looseness of the 1/K Guarantee
>
> We thank the reviewers for bringing this up. While the 1/K guarantee can be loose in some situations, we demonstrate in Lemma 8 that there exist reward functions for which the 1/K guarantee is tight. To provide a tighter guarantee, in Theorems 4 and 5, we provide lower bounds for the performance of Linear- and Shapley-Whittle dependent on the value of $\beta\_{\mathrm{linear}}$ and $\beta\_{\mathrm{shapley}}$ respectively.
>
> ### Iterative Linear-Whittle Guarantee
>
> We thank you for bringing up this question, as it is an interesting and natural question. Proof techniques used to bound the performance of the Linear- and Shapley-Whittle policies cannot be directly applied to adaptive methods because adaptive methods lack an analog for the "induced linear RMAB" or "induced Shapley RMAB". To get around this limitation, we develop new performance bounds for the iterative Linear-Whittle policy and show that a) in some scenarios, the iterative Linear-Whittle policy achieves no better than $\frac{1}{1+(K-1)\gamma}$ of the optimal average reward, and b) for any choice of reward and transitions, the iterative Linear-Whittle policy achieves at least $\frac{1}{K}$ of the optimal reward. We introduce formal theorem statements and proof sketches in the global rebuttal.
>
> ### Theorem 3 Significance
>
> We thank the reviewers for bringing up the purpose of Theorem 3. We agree with the reviewer that Theorem 3 does not guarantee the long-term reward is bounded. Theorem 3 demonstrates that it is possible to achieve a 1-1/e approximation to the Q value, which serves as an initial attempt at solving the problem. However, in addition to the algorithm not guaranteeing a long-term bound on the reward, such a method is computationally intractable, as it requires exponential time to learn such a Q function. Theorem 3 motivates the need for computationally feasible algorithms with long-term reward guarantees, as initial reinforcement learning-based solutions fail to solve the problem. We plan to clarify this further in the camera-ready version of our manuscript.
>
> ### Example 4.1 Approximation Ratio
>
> In Example 4.1, we show that our Linear-Whittle policy achieves a 1/2 approximation. We demonstrate this by first computing an upper bound on the optimal reward; we do so by upper bounding the individual contribution from each arm ($p\_{i}(s\_{i})a\_{i})$), then maximizing this by summing over arms ($\sum\_{i=1}^{N} p\_{i}(s\_{i})a\_{i})$). This allows us to find that $\mathrm{OPT} \leq 6$ . We then compute the arms which are pulled by the Linear-Whittle policy, and find that the reward for this is 3. Therefore, the Linear-Whittle policy achieves at least 1/2 of the optimal reward. We plan to add more details to our analysis in Example 4.1, and include these details in the paper.
>
> ### Explaining Iterative Linear-Whittle
>
> We thank the reviewers for pointing this out, and the reviewer is correct that $X$ are the decisions from previous iterations. To improve clarity, we plan on including pseudocode in the camera-ready version of the manuscript.
>
> ### Explaining MCTS with iterative algorithms
>
> We include details on our MCTS algorithm in Algorithm 1 of the Appendix, but plan to incorporate this into the main paper and include further details. At its core, our MCTS algorithm computes a variant of the Linear-Whittle index for different actions. Each node in MCTS represents a partial selection of arms, and children nodes indicate new possible arms to be selected. During a rollout, we select additional arms to pull for the current timestep, and after this, we have a candidate set of arms to pull corresponding to an action $\mathbf{a}$. We can compute the exact global and local reward using this and use index-based approaches to estimate the future reward for the action $\mathbf{a}$. We combine these two values to assess a total reward associated with the action $\mathbf{a}$, and repeat this procedure across different combinations of arms.

---

> > ### Comment · Reviewer_t2ns · 2024-08-11
> >
> > Thank you for your response! I think I will keep my current score. Regarding Example 4.1, your analysis seems to assume on a=[1,1,0,0] and s=[1,1,1,1]. By (4), shouldn't you consider all a and s, not just this pair?

---

> > > ### Author Response · Authors · 2024-08-11
> > > **Thank you for your response**
> > >
> > > Thank you for your comment! To answer your question, for brevity, we displayed the minimizing pair of a and s (which are a=[1,1,0,0] and s=[1,1,1,1]), but you are correct in that all pairs a and s should be considered to compute $\beta$. We plan to update our manuscript to further specify and clarify this.

---

### Official Review · Reviewer_oZCz · 2024-07-08

**Soundness:** 3
**Presentation:** 2
**Contribution:** 2
**Rating:** 4
**Confidence:** 4

**Summary:**

Conventional RMAB problems model the rewards with respect to individual arms. Authors claim that for some instances (such as food rescue), a global non-separable reward function exists, because of which solutions to RMAB cannot be applied to such problem.
To address this NP-hard problem, authors modify Whittle index policy for linear cases, and propose two adaptive policies for nonlinear cases.
Theoretical results (competitive ratio) are provided for modified index policy, and experiments show the performance of proposed algorithms.

**Strengths:**

This work opens a new topic that hasn’t been explored in RMAB, where there is a global utility function.

The authors provide theoretical bounds for linear and shapley whittle index policies.

The experiment part explores different combinations of policies and reward functions. Both synthetic and real traces are simulated to support the superior of proposed algorithms.

**Weaknesses:**

1. The major weakness I’m thinking of is the model itself. I might need more input from the authors to clarify why RMAB-G is required as a general model, instead of some specific scenario. The authors start with food rescue problems, that the platform wants to maximize the probability of one rescue task completion, while maintaining high engaging levels.

This model makes sense to me, but when I’m trying to understand other examples, such as peer review, I cannot find a good explanation for the local arm reward. The state of arm(reviewer) here is if they are available, so what will be the local arm reward? Maximize the availability of the reviewers?

The authors have mentioned several other applications for RMAB-G such as blood donation, emergency dispatch, could you clarify how these applications model global and local rewards?

Can the authors provide a general requirement for the models that need RMAB-G?

2. I’m also having a hard time understanding some of the global reward settings in section 6.1. More specifically the necessity of global reward function.
For example, for linear reward function, if we set the individual reward for each arm as a very naïve m_i s_i a_i + s_i, the solution for maximizing this reward for each arm seems to generate the largest global reward function? The same holds for max global reward, that we can just set the local reward as m_i s_i a_i.

3. Another question that I have is, it seems that MCTS requires local searching, which is a relatively complicated algorithm. Given known transition probabilities, Whittle index can be computed (or approximated), then rest of the algorithm is just looking up the index tables. I understand that proposed algorithms have better reward performance, but what is the time/ complexity sacrifice?

**Questions:**

Please refer to weakness for my major concerns.


Some minor questions:

1. Line 84, Shouldn’t it be R(s,a) = R_glob plus \sum R_i instead of times?

2. Line 103, To clarify, Whittle index policy is optimal in asymptotic regime.

3. For experiments, how many time slots (food rescue tasks) are deployed to measure the average reward? Also have you duplicated arms for the experiments, or the proposed algorithms can achieve very close to optimal reward even for 4 different arms?

I’m having this question because based on my previous experience with RMABs, Whittle index may perform a bit far from optimal if there is only 1 duplicate for each type of arm. This question also corresponds to previous question that index policies are optimal in asymptotic regime.

**Limitations:**

My scores are based on the concern of model itself, as well as the necessity of global reward function in some circumstances. If my statement is wrong or inaccurate, I’m willing to adjust the scores accordingly.

---

> ### Author Rebuttal · Authors · 2024-08-06
>
> Dear Reviewer oZCz,
>
> We thank you for your suggestions and these comments will help improve our final manuscript. We appreciate that you view our work as “opening a new topic” and are glad that you recognize our theoretical bounds. Additionally, we are happy that you appreciated our empirical results and found that they supported our proposed algorithms. We provide answers to each of your questions below:
>
> ### RMAB-G Model in Peer Review
>
> We thank the authors for bringing up their questions about applying RMAB-G to the peer review setting. If peer review institutions focus exclusively on review quality, then the local reward could be 0. Such a scenario is allowed within our model and all performance guarantees still apply to such a situation. We empirically analyze situations with zero local reward in Appendix I.
>
> ### Global and Local Rewards in Blood Donation and Emergency Dispatch
>
> The volunteer emergency dispatch situation parallels the food rescue scenario for both the global and local reward. In volunteer emergency dispatch, volunteers are notified about emergencies, to have volunteers assist with an emergency (an example of this can be seen through the app PulsePoint). The global reward is the probability that any volunteer helps out with an emergency, which corresponds to the probability reward. The local reward corresponds to engagement; similar to food rescue, emergency dispatch platforms want to ensure high engagement rates from their volunteers.
>
> In blood donation, the objective is to match blood donors to blood donation opportunities [1]. Local rewards in this situation correspond to the number of active blood donors. Global rewards could refer to the total amount of blood donated weighted by some fairness constraint proportional to the squared difference between rural and non-rural hospitals [1]. Alternatively, there could be other domain-specific considerations that play into the choice of local and global rewards.
>
> ### General Model Requirements
>
> To apply our proposed algorithms, we require two things of the reward function: a) it is submodular in the arms pulled, meaning that the marginal gain for pulling an arm is diminished as additional arms are pulled simultaneously, and b) it is monotonically increasing in the arms pulled (pulling more arms cannot decrease the reward). Submodular monotonic reward functions are common assumptions for set functions and arise naturally in many situations including explainability [2], reinforcement learning [3], and economics [4].
>
> ### Global Rewards for Maximization
>
> We thank the reviewer for this question, and plan to include further details on the maximization reward in the paper. We agree with the reviewer that for the linear global reward, naively applying the proposed solution is optimal. However, this is not true for the maximization global reward.
>
> For example, suppose that we have three arms ($N=3$) with a budget of two ($K=2$). Suppose that the global reward is the maximization reward, while the local rewards correspond to whether arms are in state 1: $R\_{i}(s\_{i},a\_{i}) = s\_{i}$. Suppose that $m\_{1} = 5$, while $m\_{2} = 4$ and $m\_{3} = 2$. Finally, suppose that arms 1 and 2 remain in state $s\_{1}=1, s\_{2}=1$, while arm 3 is in state 1 if pulled ($P\_{3}(1,1,1) = 1$). In this scenario, naively maximizing $m\_{i} s\_{i} a\_{i} + s\_{i}$ results in pulling arms 1 and 2, while the optimal set of arms to pull is 1 and 3. Neither pulling arm 2 nor arm 3 provides any benefit to the global reward when arm 1 is already pulled. However, pulling arm 3 improves the local reward more than pulling arm 2, so pulling arms 1 and 3 is optimal. More generally, because the maximum function is non-linear and non-separable, approximating the reward purely linearly can result in poor performance due to overestimation. We demonstrate this empirically in Figure 2, where our Linear-Whittle policy, which estimates the reward in a similar manner, performs worse than our Shapley-Whittle policy, which uses Shapley values to estimate the reward function.
>
> ### Time Tradeoff for MCTS
>
> We agree with the reviewer that the time complexity of MCTS is an important consideration when deciding which algorithm to use. We agree that MCTS could run slowly due to the effect of local searching; we quantify this in Figure 3, where we plot how the time needed to run MCTS varies with the problem size. In practice, when N<=100, the MCTS-based algorithms can run in under 30 seconds, which is suitable for real-world applications such as food rescue (which require that algorithms run in under 1 minute).
>
> ### Experimental details for food rescue
>
> We thank the reviewer for this question. We provide details on the number of food rescue slots in Appendix A, and plan to bring this forward into the main paper. For all experiments, we run 50 food rescue trips, and average this across 15 seeds, and 5 trials per seed. We select T=50, because when using the discounted reward, $\gamma^50 < 0.005$, and so the rewards beyond T=50 are relatively small.
>
> ### Smaller comments
>
> We thank the reviewers for their writing suggestions, and we plan to change the typo on line 84 and correct line 103 so it specifies optimality in the asymptotic regime.
>
> ### References
>
> [1] McElfresh, Duncan C., et al. "Matching algorithms for blood donation." Proceedings of the 21st ACM Conference on Economics and Computation. 2020.
>
> [2] Chen, Ruoyu, et al. "Less is more: Fewer interpretable region via submodular subset selection." arXiv preprint arXiv:2402.09164 (2024).
>
> [3] Prajapat, Manish, et al. "Submodular reinforcement learning." arXiv preprint arXiv:2307.13372 (2023).
>
> [4] Chateauneuf, Alain, and Bernard Cornet. "Submodular financial markets with frictions." Economic Theory 73.2 (2022): 721-744.

---

### Official Review · Reviewer_TFrk · 2024-07-19

**Soundness:** 4
**Presentation:** 3
**Contribution:** 4
**Rating:** 7
**Confidence:** 4

**Summary:**

This paper studies the popular Restless Multi-Armed Bandit (RMAB) problem and aims to tackle a key limitation – which is that, in RMABs, the rewards are assumed to be separable across arms. This is a limitation because in many scenarios, the overall reward may not simply be a sum over individual rewards of arms, but these rewards may be tied inextricably with each other.

To tackle this issue, the paper proposed the RMAB-G framework with non-separable global rewards. The paper shows hardness results on the RMAB-G problem. Further, the paper proposes index-based policies, similar to the original “whittle index” policy for regular RMABs – the two indexes proposed are the “linear whittle” and the “shapely whittle” indexes.

The paper also proves approximation bounds on the performance of these indexes and carries out empirical studies based on synthetic and real-world data to demonstrate the good performance.

**Strengths:**

1. I believe a key strength of the paper is proposing the RMAB-G model with a notion of non-separable global rewards. While RMABs have been extensively studied before, this formulation (and its solution) is novel to the best of my knowledge.

2. THe proposed linear and shapely whittle indexes intuitively make sense. I quite like the concept and augmentations made to the original whittle index to solve the RMAB-G problem.

3. The results presented in the paper are grounded in theory – the paper provides approximation bounds for the proposed solutions. It also proves other minor theoretical results such as hardness of the RMAB-G.

4. Empirical results look good: there are experimental results on both synthetic as well as real-world data.

**Weaknesses:**

1. Motivation: While the new RMAB-G framework and the provided analysis is interesting from a technical pov, the motivation for this setup / application to food rescue seems a little tortured. For instance, in line 66-67, the paper says that global rewards are needed because we cannot split the reward into per-volunteer functions. However, I’m not sure why — isn’t the total reward simply the sum of probabilities of individuals carrying our their assigned rescue tasks?

2. Scalability: My worry is that the proposed solutions may not scale well. The experiments are all run on a small number of arms. Technically, the bottleneck might come from computing the shapely index itself. It seems like an expensive step with a min over exponential number of state vectors.

3. Both the proposed index policies seem to depend on the budget K. Contrary to the regular whittle index, which is budget-agnostic, this seems a little less clean (and also perhaps a hurdle?). Is it possible that if the budget changes from K to K+1, the selected arms may change drastically because all the index values of all arms changed?

4. Some concepts are not introduced in the paper. For instance, indexability, whittle index, etc. are assumed to be common knowledge, potentially making the paper difficult to access for someone not familiar with these concepts.

**Questions:**

1. In line 54, why is whittle index defined as the min w where Q(., 0) > Q(., 1)? As far as I know whittle index is defined as the point where the two Q values become equal?
2. Are the assumptions on the reward functions (monotonic, submodular, etc.) true for the food rescue setting or the blood donation example?
3. Line 86: Is the RHS missing a +?
4. In Theorem 3, what does it mean for the function g(s) to be submodular in s? Why is this a reasonable assumption?
5. In the expression for u(s_i) in line 117, what do factorial terms stand for?
6. In computing the shapely index, how stable is the index wrt K? For example if K were to increase by 1, is it possible that the top-K arms are significantly different from the top-(K+1) arms because all the indices changed?
7. In line 123: why is it true that “this approach could lead to more accurate estimates”?
8. In Theorem 4, in the expression for \beta_{linear}, how is the R(s,a) term determined?
9. In line 230, what is the difference between a trial and a seed?
10. In first paragraph the paper cites maternal health [6] but this paper seems to have nothing to do with maternal health. Perhaps the authors wanted to cite the following:
Aditya Mate, Lovish Madaan, Aparna Taneja, Neha Madhiwalla, Shresth Verma, Gargi Singh, Aparna Hegde, Pradeep Varakantham and Milind Tambe.
“Field Study in Deploying Restless Multi-Armed Bandits: Assisting Non-Profits in Improving Maternal and Child Health”

**Limitations:**

yes

---

> ### Author Rebuttal · Authors · 2024-08-06
>
> Dear Reviewer TFrk,
>
> We thank you for your suggestions and insights, and your comments will help improve our final manuscript. We are happy that you find our study novel, and find the RMAB-G model to be a key strength of our paper. Additionally, we are glad that you find our policies intuitive and grounded in theory. Finally, we appreciate that you think our empirical results are good, both for synthetic and real-world experiments. We provide answers to each of your questions below:
>
> ### Motivation for Food Rescue Scenario
>
> We thank the reviewer for bringing up this important question. We note that the probability of any volunteer matching to a rescue trip is not simply the sum of their match probabilities. For example, if there are 2 volunteers, each with a match probability of 1/2, then the probability of any volunteer matching is 3/4, rather than 1/2+1/2 = 1. This is because we compute the probability that no volunteer successfully matches to a trip, which is a nonlinear function of the individual match probabilities.
>
> ### Scalability of Method
>
> We agree with the reviewer that the scalability of our methods is important to assess the real-world applicability of our methods. In Appendix I and Appendix J, we vary the number of arms between 25 and 1000, and compare the performance of our policies. Index-based policies run quickly because the indices can be pre-computed and Shapley values can also be computed quickly because we estimate Shapley values using a subset of arm combinations. While adaptive policies, such as MCTS, run slower than index-based policies, when N <= 50, both adaptive and index-based policies can be computed in under 15 seconds, which is fast enough for real-world use cases such as food rescue. For large N, we can use index-based methods, which run in under a second per timestep for N=1000.
>
> ### Impact of Budget on Arms Selected
>
> We thank the reviewer for bringing up an interesting question about how the budget impacts the actions chosen by our index-based policies. For the Linear-Whittle policy, all arms that are selected with a budget of $K$ are selected for a budget of $K+1$ as well (within a particular timestep), because $p\_{i}(s\_{i})$ is independent of the budget. However, for the Shapley-Whittle policy, the impact of the budget on the arms selected is more complicated because $u\_{i}(s\_{i})$ is not independent of the budget. We do not believe that the Shapley-Whittle indices should change drastically when increasing the budget because the Shapley values are computed across many combinations of arms. Intuitively, increasing the budget should change the size of each arm combination, but the average reward of the arm combinations should not change drastically.
>
> ### Introducing Indexability and Whittle Index
>
> We agree with the reviewer that indexability and the Whittle index are important concepts that can help readers better understand our work. In the camera-ready version of our work, we plan to include additional background on these concepts to make our paper more accessible.
>
> ### Assumptions for Food Rescue and Blood Donation
>
> We thank the reviewer for bringing up an interesting question about our reward functions in food rescue and blood donation. As mentioned in Section 3, we model the global reward in food rescue using the probability reward function, which is both monotonic and submodular. A similar reward function can be used for the blood donation setting, as notifying donors about donation opportunities can only result in more blood being donated. Additionally, notifying additional donors leads to diminishing returns, due to capacity restrictions on the total amount of blood donated.
>
> ### Submodular g function
>
> Theorem 3 assumes $g(\mathbf{s})$ is submodular in $\mathbf{s}$. $g(\mathbf{s})$ represents the maximum reward attainable from state $\mathbf{s}$. If $g(\mathbf{s})$ is submodular, then additional arms in the 1 state leads to diminishing marginal returns. Such an assumption holds true across each of the reward functions used in our experiments, and is commonly seen when additional present arms result in diminishing returns. We plan to include a brief discussion of the meaning of the assumptions for Theorem 3.
>
> ### Better estimates from Shapley-Whittle
>
> We thank the reviewers for pointing this out. On line 123, we state the Shapley-Whittle policy could lead to better estimates of the reward when compared to the Linear-Whittle policy. The Shapley-Whittle policy uses Shapley values to estimate the marginal contribution of each arm, while the Linear-Whittle policy overestimates marginal contributions. As a result, in many cases, the Shapley-Whittle index provides a better estimate of the reward because marginal contributions are averaged across many combinations of arms. This intuition is backed up by empirical evidence, where Shapley-based policies perform better than Linear-based policies. We plan to update our write-up to make this point clearer.
>
> ### Other Questions
>
> We thank the reviewer for pointing out our typo on Line 86, and we additionally plan to change Line 54 to Q(.,0) = Q(.,1).
>
> On line 117, the factorial terms arise from the original definition of Shapley values; these terms account for the number of orderings of arms for a particular action combination.
>
> For Theorem 4, $R(\mathbf{s},\mathbf{a})$ term can be computed for any given $\mathbf{s}$ and $\mathbf{a}$. $\beta\_{\mathrm{linear}}$ can then be computed by minimizing this ratio across all choices for the action and state.
>
> When we vary the trial, the starting state, $\mathbf{s}^{{0)}$ changes, while the transition matrices, $\mathcal{P}$, and reward parameters remain constant. However, when we change the seed, the starting state, transition matrices, and reward parameters all change.
>
> We thank the reviewer for pointing out our misplaced citation, and we will fix this in the camera-ready.

---

### Author Rebuttal · Authors · 2024-08-06

We thank the reviewers for their insightful comments and for taking the time to carefully read through our work. We are pleased that reviewers find our problem formulation novel (reviewers TFrk18, oZCz07, and t2ns05) and motivated by real-world applications (reviewer a47J02). We are happy to see that reviewers find our proposed policies intuitive (reviewer TFrk18) with good empirical (reviewers TFrk18, oZCz07, and a47J02) and theoretical (reviewers TFrk18, oZCz07, t2ns05, and a47J02) backing. We additionally appreciate that reviewers find our set of empirical evaluations comprehensive (reviewer a47J02). Your comments have greatly helped improve our work.

We hope to address your comments and questions in this rebuttal. We first describe a set of new approximation bounds for adaptive methods, which was discussed by multiple reviewers, while we reply individually to each reviewer for any comments they might have.

# Summary of New Additions

## Approximation Bounds for Adaptive Methods

Following feedback from reviewers a47J and t2ns, we develop performance guarantees for adaptive methods. Proof techniques used to bound the performance of the Linear- and Shapley-Whittle policies cannot be directly applied to adaptive methods because adaptive methods lack an analog for the "induced linear RMAB" or "induced Shapley RMAB".

To get around this limitation, we develop new performance bounds for the iterative Linear-Whittle policy and show that a) in some scenarios, the iterative Linear-Whittle policy achieves no better than $\frac{1}{1+(K-1)\gamma}$ of the optimal average reward, and b) for any choice of reward and transitions, the iterative Linear-Whittle policy achieves at least $\frac{1}{K}$ of the optimal reward. We formally state these theorems and provide proof sketches below.

Theorem 1:     Let $\pi\_{\mathrm{IL}}$ be the iterative linear policy.  Let $\mathrm{ALG} = \mathbb{E}\_{(\mathbf{s},\mathbf{a})\sim (P,\pi\_{\mathrm{IL}})}[\sum\_{t=0}^{\infty}  \gamma^{t} R(\mathbf{s},\mathbf{a})]$ and $\mathrm{OPT} = \max\limits\_{\pi} \mathbb{E}\_{(\mathbf{s},\mathbf{a})\sim (P,\pi)}[\sum\_{t=0}^{\infty} \gamma^{t}  R(\mathbf{s},\mathbf{a})]$. Then there exists a reward function, $R(\mathbf{s},\mathbf{a})$, and transition probabilities, $\mathcal{P}$, so $\mathrm{ALG} = \mathrm{OPT} \frac{1}{1+(K-1) \gamma}$.

Theorem 2:     For any fixed set of transitions, $\mathcal{P}$, let $\mathrm{OPT} = \max\limits\_{\pi} \mathbb{E}\_{(\mathbf{s},\mathbf{a})\sim (P,\pi)}[\frac{1}{T} \sum_{t=0}^{T-1}  R(\mathbf{s},\mathbf{a})]$ for some $T$.
    For an \abr{rmab-g} $(\mathcal{S},\mathcal{A},R\_{i},R\_{\mathrm{glob}},P_{i},\gamma)$, let $R'\_{i}(s\_{i},a\_{i}) = R\_{i}(s\_{i},a\_{i}) + p\_{i}(s_{i}) a\_{i}$, and let the induced linear \abr{rmab} be $(\mathcal{S},\mathcal{A},R'\_{i},P\_{i},\gamma)$.
    Let $\pi\_{\mathrm{IL}}$ be the iterative Linear-Whittle policy, and let $\mathrm{ALG} = \mathbb{E}\_{(\mathbf{s},\mathbf{a})\sim (P,\pi\_{\mathrm{IL}})}[\frac{1}{T} \sum\_{t=0}^{T-1}  R(\mathbf{s},\mathbf{a})]$.
    For any policy, $\pi$, let $\pi\_{i,t}$ be the augmented policy that pulls arms according to $\pi$ and additionally pulls arm $i$ at timestep $t$.
    If $\mathbb{E}\_{(\mathbf{s},\mathbf{a})\sim (P,\pi\_{i,t})}[\frac{1}{T} \sum\_{t=0}^{T-1}  R(\mathbf{s},\mathbf{a})] \geq \mathbb{E}\_{(\mathbf{s},\mathbf{a})\sim (P,\pi)}[\frac{1}{T} \sum\_{t=0}^{T-1}  R(\mathbf{s},\mathbf{a})]$ and the induced linear \abr{rmab} is irreducible and indexable with the uniform global attractor property, then $\mathrm{ALG} \geq \frac{1}{K} \mathrm{OPT}$ asymptotically in $N$ for any set of transitions, $\mathcal{P}$.

Proof sketch of Theorem 1: To demonstrate an upper bound on the performance of the iterative Linear-Whittle policy, we construct a reward function with $2K-1$ arms. The optimal policy always pulls arms $1$ and $K+1,\ldots,2K-1$, while the iterative Linear-Whittle policy pulls arms $1,\ldots,K$. We construct the reward function in such a way that the reward of pulling arms $1,\ldots,K$ is $1$ for each timestep, while pulling arms $1$ and $K+1,\ldots,2K-1$ results in a reward of $1 + (K-1) \gamma$ per timestep.

Proof sketch of Theorem 2: To demonstrate a lower bound on the performance of the iterative Linear-Whittle policy, we first demonstrate that the iterative Linear-Whittle policy with budget $K$ does no worse than the Linear-Whittle policy with budget $1$. We then show that the Linear-Whittle policy with budget $1$ is at least a $\frac{1}{K}$ of the optimal reward. Taken together, this implies that the iterative Linear-Whittle policy achieves at least $\frac{1}{K}$ of the optimal reward.

---

### Decision · Program_Chairs · 2024-09-25

**Decision:**

Accept (poster)

**Comment:**

The paper is acknowledged to make several foundational contributions in the theory and algorithms for restless bandits with non-additive ('global') reward structure across the arms.

There were some concerns raised by some of the referees about the applicability of the problem to real world settings. The author response issued clarifications, which have been helpful in understanding the importance of modeling nonlinear, global rewards in several applications depending on the underlying objective.

In view of the largely favorable views expressed in the referees and the strengths of the submission, the paper is recommended for acceptance.